# Lighting up metal nanoclusters by the H$_2$O-dictated electron relaxation dynamics

Yuan Zhong [1,7], Xue Wang [1,7], Zhou Huang[2,7], Yao Wei [3,7], Qing Tang [2] ✉, Songqi Gu [3] ✉, Tingting Li[4], Weinan Dong[1], Feng Jiang[1], Haifeng Zhu[1], Yujia Shi[5], Zhi Zhou [6], Yu Zhang [1] ✉, Xue Bai [1] ✉ & Zhennan Wu [1] ✉

The modulation of traps has found attractive attention to optimize the performance of luminescent materials, while the understanding of trap-involved photoluminescence management of metal nanoclusters greatly lags behind, thus extensively impeding their increasing acceptance as the promising chromophores. Here, we report an efficient passivation of the structural oxygen vacancies in AuAg nanoclusters by leveraging the H$_2$O molecules, achieving a sensitive color tuning from 536 to 480 nm and remarkably boosting photoluminescence quantum yield from 5.3% (trap-state emission) to 91.6% (native-state emission). In detail, favored electron transfer relevant to the structural oxygen vacancies of AuAg nanoclusters contributes to the weak trap-state emission, which is capable of being restrained by the H$_2$O molecules by taking Au-O and Ag-O bonds. This scenario allows the dominated native-state emission with a faster radiative rate. In parallel, the H$_2$O molecules can rigidify the landscape of AuAg nanoclusters leveraging on the hydrogen bonding, thus enabling an efficient suppression of electron-optical phonon coupling with a decelerated non-radiative rate. The presented study deepens the understanding of tailoring the photoluminescence properties of metal nanoclusters by manipulating surface trap chemistry and electron relaxation dynamics, which would shed new light on luminescent metal nanoclusters with customizable performance.

Traps play a key role in modulating the optical and electronic properties of materials, which has added their acceptance in various sectors of applications, including energy storage, catalysis, photovoltaics, optoelectronics, etc[1–4]. Traps, including vacancies, interstitials, and substitutional impurities, are often generated across multiple processes from synthesis, purification, modification, and storage to applications[5]. Especially, for luminescent materials, structural traps in general introduce intermediate energy levels, which can act as electron traps or modify electron transfer pathways, and thereby affecting excited-state electron relaxation dynamics[6,7]. Such perturbations can significantly influence radiative and non-radiative relaxations, and hence photoluminescence (PL) performance. In this vein, the controlled introduction of structural traps (i.e., trap engineering) is an efficient strategy for tailoring the optical properties of materials[8]. The

[1]State Key Laboratory of Integrated Optoelectronics, College of Electronic Science and Engineering, Jilin University, Changchun, P. R. China. [2]Chongqing Key Laboratory of Chemical Theory and Mechanism, School of Chemistry and Chemical Engineering, Chongqing University, Chongqing, P. R. China. [3]Shanghai Advanced Research Institute, Chinese Academy of Sciences, Shanghai, P. R. China. [4]College of Materials Science and Engineering, Jilin Jianzhu University, Changchun, P. R. China. [5]Department of Oral Implantology, Jilin Provincial Key Laboratory of Sciences and Technology for Stomatology Nanoengineering, School and Hospital of Stomatology, Jilin University, Changchun, P. R. China. [6]Hunan Optical Agriculture Engineering Technology Research Center, School of Chemistry and Materials Science, Hunan Agricultural University, Changsha, P. R. China. [7]These authors contributed equally: Yuan Zhong, Xue Wang, Zhou Huang, Yao Wei. ✉e-mail: qingtang@cqu.edu.cn; gusq@sari.ac.cn; yuzhang@jlu.edu.cn; baix@jlu.edu.cn; wuzn@jlu.edu.cn

rational manipulation of trap concentrations enables optimized PL quantum yield (PLQY) and diversified functionalities, as reported in luminescent materials like ZnO and CdSeTe quantum dots (QDs), and Eu-doped inorganic phosphors[9–11], where trap states enhance radiative recombination, prolong PL lifetimes, and generate broad-band white emission. Therefore, structural traps are crucial in the precise modulation of PL property and hold great potential for the ideal customization of emission performance.

Metal nanoclusters (NCs) are composed of monolayer-ligand-protected several to a few hundred metal-atom aggregates, which can be well-described by a "divide-and-protect" model[12,13]. The integration of inorganic metal core and organic-inorganic staple motifs makes them be regarded as "metallic molecules" and exhibit diverse molecule-like properties[14–21]. In particular, metal NCs display a strong dependence on structural defects due to their ultra-small size (typically less than 3 nm) and abundance of dangling bonds and/or exposed surfaces[22–26]. The dynamic landscape evolution and topological complexity, combined with a high crystalline index, further drive the formation of structural traps[27–30]. In addition, the transitional interaction between the metal core and protecting ligands would induce strain within the NCs, promoting the development of structural traps[31,32]. Recent reports have manifested the discoveries of surface metal vacancies and point defects in the crystal structures of metal NCs[33–35]. And, as their single metal-atom or single ligand-molecule dependent chemical and physical properties, these structural traps were demonstrated to serve a critical role in modulating their properties[36–43]. However, on the one hand, the species of metal NCs containing structural traps and their trap-induced property optimizations have not yet been sufficiently investigated so far, mainly due to the fact that the formation of structural traps reduces the stability of metal NCs, allowing them to be characterized only in single crystals or in solutions[44]. On the other hand, the rare reports on the trap-tolerant metal NCs are severely limited in illustrating their changes in topology[45,46]. No special emphasis has been put into clarifying the impact and corresponding mechanism of the manipulation of surface trap chemistry and electron relaxation dynamics on their PL properties (e.g., trap-state emission, and electron transfer between multiple luminescent centers)[47–49]. It, therefore, becomes an urgent task to establish a clear relationship between structural traps and PL properties to promote the development of luminescent metal NCs with tunable emitting color and ultrahigh PLQY.

Herein, we report the passivation of the trap state with $H_2O$ molecules achieves the controlled emitting color from 536 nm-green to 480 nm-sky-blue and significantly boosted PLQY from 5.3% to 91.6% in AuAg NCs. Specifically, the freeze-dried AuAg NCs feature structural oxygen vacancies and thus give 536 nm trap-state emission. The spontaneous absorption of $H_2O$ molecules in air would form Au-O and Ag-O bonds to passivate such structural oxygen vacancies, which is beneficial to block the electron transfer from the native state to the trap one and thereby switching the predominant radiative relaxation channel to the 480 nm emission of AuAg NCs. In addition, the absorbed $H_2O$ molecules can also interact with the protecting ligands through hydrogen bonds. The rigidified landscape in AuAg NCs reduces the strength and energy of non-radiative electron-optical phonon coupling in staple motifs. As a result, the ultrahigh PLQY of 91.6% can be realized in the aqueous solution of AuAg NCs at room temperature. Our findings not only deepen the understanding of trap-state emission in metal NCs but also provide a simple and effective strategy to passivate structural traps towards significant enhancement in emission intensity and stability adding to their acceptance in diverse sectors of practical applications.

## Results

### $H_2O$-trigged PL tuning in AuAg NCs

The aqueous solution of 3-mercaptopropionic acid (MPA) ligands-protected AuAg NCs was first synthesized (denoted as AuAg-S NCs, see Methods Section). Surprisingly, we found the freeze-dried AuAg NCs (denoted as AuAg-D NCs) powder can quickly (within 1 s) adsorb $H_2O$ molecules in 56% RH air to transform into hydrated AuAg NCs (denoted as AuAg-H NCs, which specifically refers to AuAg-D NCs aged with 56% RH). The corresponding body color of AuAg-D, AuAg-H, and AuAg-S NCs changed from deep yellow to light green. Under 365 nm near-UV light illumination, AuAg-D NCs emit faint green light while AuAg-H and AuAg-S NCs show very bright sky-blue emission (Fig. 1a). This $H_2O$-triggered significant changes in the PL color and intensity of AuAg NCs is observed and reported for the first time for luminescent metal NCs. To correlate the PL property of AuAg NCs with the surrounding RH condition, we have developed a homemade RH-dependent PL testing system on a FLS1000 spectrofluorometer (Fig. 1b). The RH condition around AuAg-D NCs in powder cuvette can be controlled in the range of 2–56% by regulating the flow rate of the air and the dry $N_2$ gas stream. As shown in Fig. 1c, upon 365 nm light excitation, AuAg-D NCs give a 536 nm green emission band with a full width at half maximum (FWHM) of 223.5 meV (Supplementary Fig. 1 and Supplementary Table 1). The sustained absorption of $H_2O$ molecules of AuAg-D NCs would constantly blue-shift its emission band and significantly enhance the emission intensity. As a result, the terminal AuAg-S NCs show a 480 nm sky-blue emission with the narrowest FWHM of 130.5 meV for luminescent metal NCs so far. Accordingly, the Commission Internationale de L'Eclairage (CIE) chromaticity coordinate of AuAg NCs can be tuned in an extensive range, from (0.23, 0.67) for AuAg-D NCs to (0.10, 0.17) for AuAg-S NCs (Fig. 1d). The bandgap energies ($E_g$) of AuAg-D, AuAg-H, and AuAg-S NCs are calculated to be 2.53, 2.63, and 2.71 eV, respectively, by Tauc plotting their corresponding absorption spectra, which indicates that the coupling of $H_2O$ molecules affect the electronic structure of AuAg NCs (Supplementary Fig. 2). The absolute PLQYs of AuAg-D, AuAg-H, and AuAg-S NCs are measured to be 6.6%, 76.7%, and 85.1%, respectively, upon 365 nm light excitation. And with near-bandgap energy excitation, their absolute PLQYs are recorded to be 5.3%, 79.5%, and 91.6%, respectively (Supplementary Figs. 3, 4). In addition, the addition of other alcohol solvents (i.e., methanol, ethylene glycol, ethanol, isopropanol, n-propanol, 1,2-propylene glycol) was found to improve the PL performance of AuAg-D NCs to varying degrees, but none of them were as pronounced as the addition of $H_2O$ (Supplementary Fig. 5). These results might be related to the higher polarity and smaller steric effect of $H_2O$ molecules.

We further carried out O $1s$ X-ray photoelectron spectroscopy (XPS) and thermogravimetric analysis (TGA) measurements to confirm the existence of $H_2O$ molecules in AuAg-H NCs powder. The subpeaks of $O_α$ (530.2 eV) and $O_β$ (531.3 eV) are found in both the O $1s$ XPS peak deconvolutions of AuAg-D and AuAg-H NCs, corresponding to -C-O and -C = O bonds in the MPA ligands, respectively (Supplementary Fig. 6). However, an additional $O_γ$ subpeak centering at 533.5 eV is found for AuAg-H NCs, suggesting that the contribution of O element from absorbed $H_2O$ molecules. In addition, comparing the TGA spectra of AuAg-D and AuAg-H NCs, a weight loss stage from room temperature to 150 °C, which is caused by the evaporation of adsorbed $H_2O$ molecules, can be clearly identified in that of AuAg-H NCs (Supplementary Fig. 7). Therefore, the significant difference in PL properties between AuAg-D and AuAg-H NCs powders is well-assigned to the adsorbed $H_2O$ molecules. Scanning electron microscope (SEM) images of AuAg-D NCs manifest that the unique water-absorbing property of AuAg-D NCs is driven by its inherently porous microstructure (Supplementary Fig. 8). The corresponding Brunauer-Emmet-Telle (BET) analysis of AuAg-D NCs further determined its mesopore diameter is ~20 nm with a BET surface area of 10.73 m²/g (Supplementary Fig. 9). In addition, AuAg-D NCs powder shows good reproducibility during the cycle of spontaneous water adsorption in 56% RH air and water desorption by pumping vacuum. The PL peaks of AuAg-D and AuAg-H NCs are nearly unchanged even

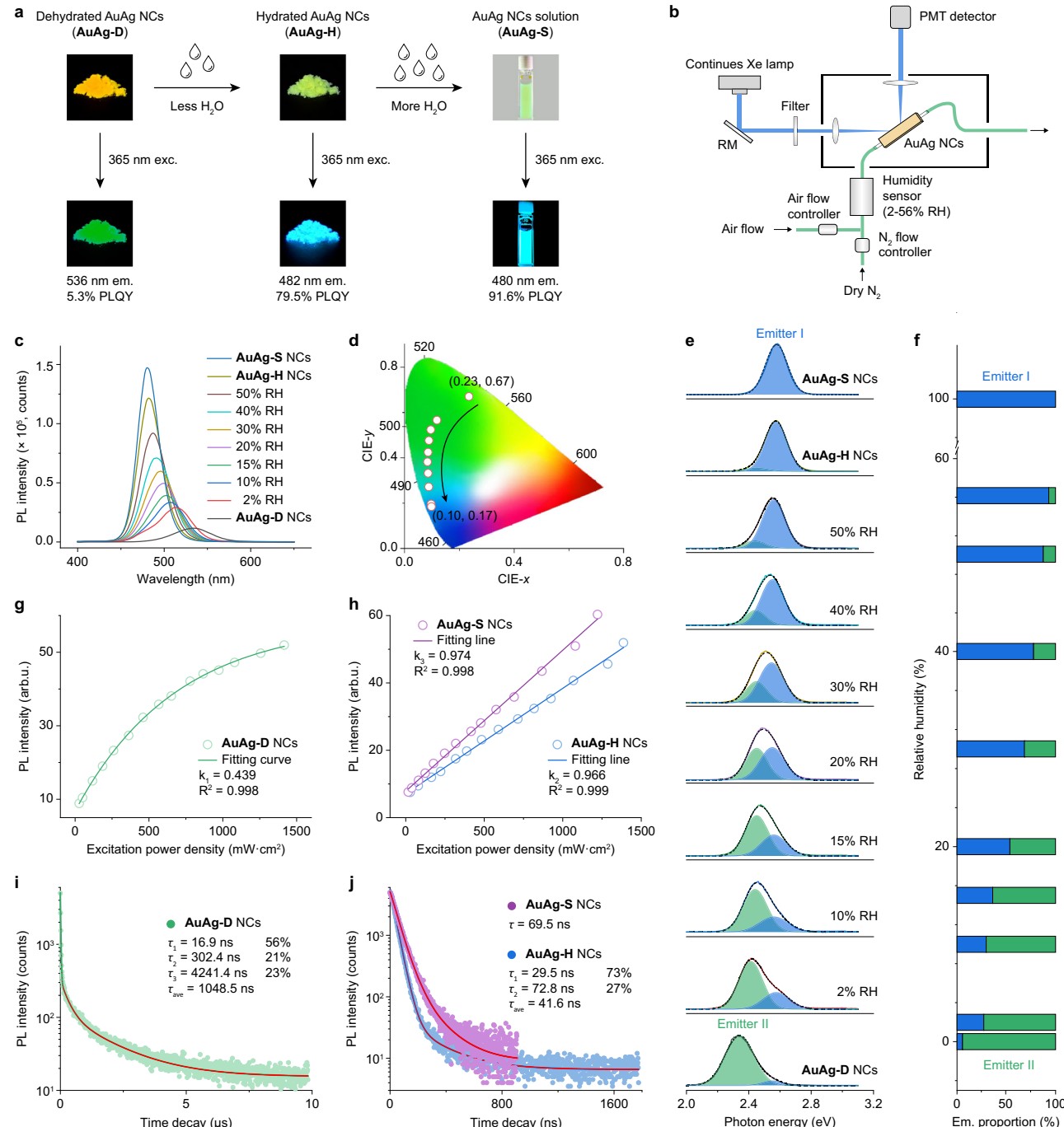

**Fig. 1 | Optical properties of AuAg NCs before and after water absorption.**
**a** Cascading evolution of AuAg-D, AuAg-H, and AuAg-S NCs, and their corresponding digital photos under sunlight (upper plane) and 365 nm light illumination (bottom plane). **b** Schematic of the homemade PL testing system with controllable RH environment around AuAg NCs (drew in Adobe Illustrator). **c** PL spectra of AuAg-D, AuAg-H, and AuAg-S NCs upon 365 nm excitation. The RH range around AuAg-D NCs is set from 2% to 56%. **d** CIE coordinate diagram of AuAg-D, AuAg-D aged in 2–56% RH environment, and AuAg-S NCs. **e** PL peak deconvolutions of AuAg-D, AuAg-D aged in

2–56% RH environment, and AuAg-S NCs. The high-energy and low-energy PL peaks are assigned to emitter I and II, respectively. **f** Variation of the proportion of emitter I and II at different RH conditions. **g**, **h** Integrated PL intensity of AuAg-D, AuAg-H, and AuAg-S NCs as a function of excitation laser power density. **i**, **j** PL decays and corresponding fitting results of AuAg-D, AuAg-H, and AuAg-S NCs. All the samples are excited by a 370 nm pulsed laser and the monitoring wavelength was set at 536 nm for AuAg-D NCs, 482 nm for AuAg-H NCs, and 480 nm for AuAg-S NCs, respectively. Source data are provided as a Source Data file.

after 100 hydration-dehydration cycles (Supplementary Fig. 10 and Supplementary Movies 1, 2).

It is worth noting that the PL profiles of AuAg-D NCs recorded in different RH conditions are not Gaussian symmetric, which means that their emission bands are jointly contributed by multiple luminescent centers. To prove this, all the PL bands are subjected to deconvolution on a photon energy scale. Emitter I (2.55 eV) and II (around 2.42 eV) can

be fitted out for all samples except the AuAg-S NCs (Fig. 1e and Supplementary Table 2). Moreover, the proportion of emitter I increased while that of emitter II declined with the increase of RH condition, suggesting that $H_2O$ molecules can change the radiative channel in AuAg NCs (Fig. 1f). Based on the extremely close PL peak between emitter I and AuAg-S NCs (2.58 eV), it is reasonable to assign emitter I in all NCs to the native emitting state of AuAg NCs with high-PLQY

sky-blue emission. On the contrary, the expression of unidentified emitter II gives low-PLQY green emission. It is, therefore, important to figure out the origin of emitter II. AuAg-H and AuAg-D NCs are then employed as the model NCs of emitters I and II for further investigations. The excitation-dependent PL spectra of AuAg-D and AuAg-H NCs confirm their PL bands are not relevant to the excitation wavelength, demonstrating that both emissions are emitted from the lowest excited-state energy level of emitter I and II (Supplementary Figs. 11, 12). Therefore, there are three possible PL origins for emitter II, namely triplet-state phosphorescence, self-trapped excitons (STEs), and trap-state emission. We first ruled out the possibility of triplet-state phosphorescence for emitter II because the PL intensity of $O_2$-saturated AuAg-D NCs is not quenched and no characteristic $^1O_2$ emission peak is detected at ~1275 nm (Supplementary Fig. 13)[50,51]. As for STEs, they typically emerge in some compounds with soft lattice and strong excitons-phonon coupling features, where the excitons can be "self-trapped" in the lattice distortions, and leading to efficient broadband emission with a large Stokes shift[52]. STEs are very sensitive to the pressure in materials because the local structural distortion is increased at high pressure and, therefore, promotes the strength of excitons-phonon coupling and transition dipole moments[53]. As a result, the PL intensity of STEs generally increased with the increase of applied pressure. As shown in the in situ high-pressure PL spectra of AuAg-D NCs, the gradually declined PL intensity at high pressure indicates that emitter II does not belong to STEs (Supplementary Fig. 14). However, the observed continuous redshift in the PL spectra of AuAg-D NCs is reminiscent of the pressure-dependent PL property of trap-state emitters whose emission depends on the trap-state depth[54]. As a proof of concept, we measured the excitation power-dependent PL spectra of AuAg-D, AuAg-H, and AuAg-S NCs under 350 nm pulsed laser pumping. With the gradually increased excitation power, all the PL intensities of AuAg-D, AuAg-H, and AuAg-S NCs enhanced without a shift in the PL position (Supplementary Fig. 15). Then, the integrated PL intensities ($I$) are subjected to fit as a function of excitation power density ($P$) by using a power law[55]:

$$I = nP^k + C \qquad (1)$$

where $n$ and $C$ represent fitting constants, and $k$ is related to the PL mechanism. In general, near band-edge excitonic PL exhibits a linear ($k$ ~ 1) power dependence on excitation power, whereas trap-state PL is expected to yield a sublinear relationship ($k < 1$) due to that the defects are fully populated with excitons at high excitation power densities[56]. The corresponding fits of AuAg-D, AuAg-H, and AuAg-S NCs give the $k$ values of 0.439, 0.966, and 0.974 (Fig. 1g, h), respectively, evidencing the emitter II in all AuAg NCs belongs to trap-state emission center.

To give more insights into the $H_2O$-trigged PL tuning in AuAg NCs, the PL lifetimes of AuAg-D, AuAg-H, and AuAg-S NCs were measured. As shown in Fig. 1i, three-time components are extracted from the PL decay of AuAg-D NCs, that is, 16.9 ($\tau_1$, 56%), 302.4 ($\tau_2$, 21%), and 4241.4 ns ($\tau_3$, 23%). The average PL lifetime ($\tau_{ave}$) of AuAg-D NCs is determined to be 1048.5 ns. However, only two-time components, namely 29.5 ($\tau_1$, 73%) and 72.8 ns ($\tau_2$, 27%), can be fitted out for the PL decay of AuAg-H NCs, with a $\tau_{ave}$ value of 41.6 ns. Furthermore, the PL decay of AuAg-S NCs shows a mono-exponential radiative decline within 69.5 ns (Fig. 1j and Supplementary Table 3). To clarify the assignment of each time component in AuAg-D and AuAg-H NCs, we carried out their temperature-dependent PL lifetime measurements in the temperature range of 10–290 K (Supplementary Fig. 16 and Supplementary Tables 4, 5). Notably, the proportion of $\tau_3$ of AuAg-D NCs remains relatively stable at all temperatures. Based on the $\mu$s-level lifetime, $\tau_3$ can be reasonably assigned to the trap-state relaxation in AuAg-D NCs. In addition, for both temperature-dependent PL lifetimes of AuAg-D and AuAg-H NCs, the proportion of $\tau_1$ declined, and that of $\tau_2$ increased with the drop in temperature. The non-radiative relaxation

in luminescent materials is caused by the phonon-assisted lattice vibrations, and it is greatly suppressed at low temperatures[57]. It is thereby rational to assign $\tau_1$ and $\tau_2$ to the non-radiative and radiative relaxation of excitons in AuAg NCs, respectively. In addition, it should be noted that the presence of $\tau_1$ and $\tau_2$ in the PL lifetime of AuAg-D NCs may be a hint that the trap-state emission originates from the native emitting state of AuAg NCs through electron transfer. From AuAg-D to AuAg-H NCs, the absence of the $\tau_3$ component suggests that the less absorbed $H_2O$ molecules can alter the radiative pathway from the trap state to the native emitting state of AuAg NCs. From AuAg-H to AuAg-S NCs, the absence of the $\tau_2$ component implies that the more absorbed $H_2O$ molecules can serve as "water ligands" to stabilize AuAg NCs and reduce non-radiative relaxation. The dual functions of $H_2O$ molecules to boost the PL property of AuAg NCs is also supported by the ~375-fold acceleration in radiative decay rate ($k_r$, which is typically hard to alter except for changes in radiative pathways) from AuAg-D to AuAg-H NCs, and ~4-fold deceleration in non-radiative decay rate ($k_{nr}$) from AuAg-H to AuAg-S NCs (Supplementary Table 6).

## Mechanism of tailoring trap-state emission with $H_2O$

To understand the mechanism of tailoring trap-state emission with $H_2O$ molecules, we need to figure out the relationship between the native emitting state of AuAg NCs and the trap emitting state. To this end, the time-resolved PL (TRPL) spectra of AuAg-D, AuAg-H, and AuAg-S NCs were measured (Supplementary Fig. 17). The PL peaks of AuAg-D and AuAg-H NCs shift from 510 to 536 nm in 48.8 ns and from 486 to 504 nm in 123.0 ns, respectively. On the contrary, the PL peak of AuAg-S NCs maintained at 480 nm in all delay times. Since both AuAg-D and AuAg-H NCs contain trap-state emission (emitter II), the continued redshifts in the PL spectra of two NCs demonstrate the underlying electron transfer process from the native emitting state of AuAg NCs to the trap emitting state. Compared to AuAg-H NCs, the broader redshift in AuAg-D NCs within a shorter delay time indicates a faster electron transfer rate (Supplementary Fig. 18). The TRPL spectra of AuAg-D, AuAg-H, and AuAg-S NCs are then subjected to wavelength-dependent PL lifetime analysis. The proportions of $\tau_3$ in AuAg-D NCs and $\tau_2$ in AuAg-H NCs keep stable before the monitoring wavelength of 530 and 490 nm, respectively. The much longer switching wavelength in AuAg-D NCs proves that the native emitting state of AuAg NCs exerts a greater impact on the trap emitting state (i.e., faster electron transfer rate, Supplementary Fig. 19 and Supplementary Table 7).

We further implemented femtosecond-streak camera and transient absorption (TA) measurements of AuAg-D, AuAg-H, and AuAg-S NCs to quantitatively evaluate the electron transfer process. As shown in Fig. 2a, the 480 nm PL traces are detected in the streak camera images of all AuAg NCs, while the 536 nm trap-state emission is detected for AuAg-D and AuAg-H NCs. Therefore, the 480 nm PL traces are extracted and the lifetime fits are applied. Interestingly, only a one-time component (>1 ns) is obtained for AuAg-S NCs, which corresponds to the radiative relaxation of the native emitting state of AuAg NCs. Whereas both the PL traces of AuAg-D and AuAg-H NCs include an ultrafast decay component (32.9 ps and 78.5 ps, respectively) and a radiative relaxation component (>1 ns, Fig. 2b). Since the electron transfer process is analogous to a non-radiative process for the electron donor, it is thereby rational to assign 32.9 ps and 78.5 ps to the electron transfer process in AuAg-D and AuAg-H NCs, respectively. Therefore, the introduced $H_2O$ molecules can suppress the electron transfer process from the native AuAg NCs emitting state to the trap-emitting state, and even completely block this process in AuAg-S NCs. We deepen these understandings by using the femtosecond-TA spectra of three AuAg NCs (Supplementary Fig. 20). As shown in the left of Fig. 2c, an obvious signal rising stage can be clearly identified within the initial 100 ps time scale, which correlates to the injection of excited-state electrons into the electron receptor (i.e., trap state) in AuAg-D NCs[58]. This electron injection process can also be decoded in

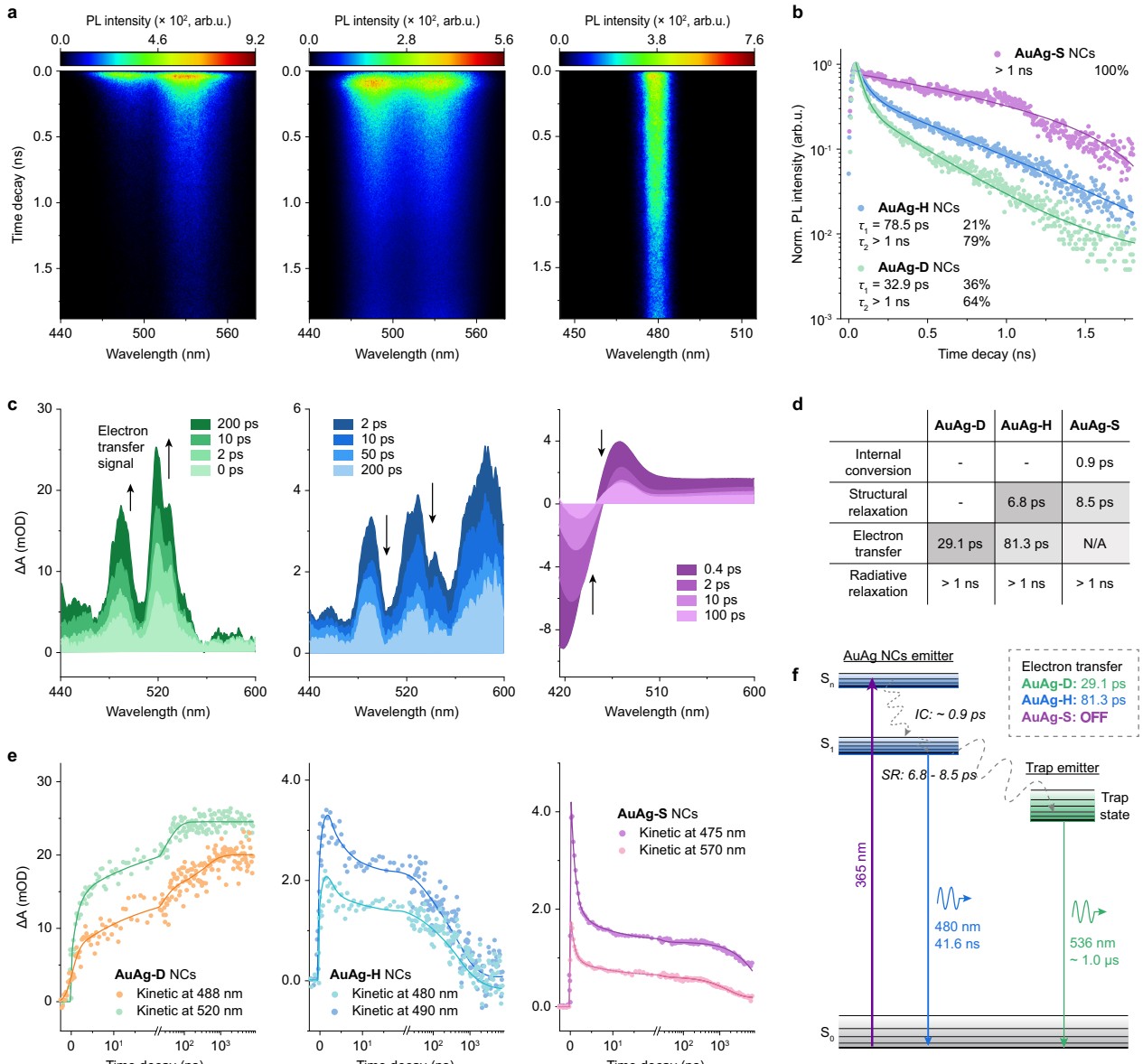

**Fig. 2 | H₂O-modulated electron transfer from native NCs state to trap state.** **a** Streak camera images of AuAg-D, AuAg-H, and AuAg-S NCs (from left to right) upon 400 nm laser pumping. **b** PL decay curves monitored at 480 nm in the streak camera images of AuAg-D, AuAg-H, and AuAg-S NCs, and their corresponding fitting results. **c** Femtosecond-TA spectra of AuAg-D, AuAg-H, and AuAg-S NCs recorded at different delay times. The scaling of the color scales is the absorption intensity, and its unit is milli-optical density (mOD). The excitation source is a 365 nm laser. **d** Time constants obtained from kinetic fittings of TA spectra at different monitoring wavelengths. **e** Decay kinetics and corresponding fittings of AuAg-D, AuAg-H, and AuAg-S NCs extracted from their femtosecond-TA maps. **f** The proposed diagram illustrates the critical mechanism of tailoring trap-state emission with water molecules. Source data are provided as a Source Data file.

the short-time TRPL spectra of AuAg-D NCs, especially shifting the monitoring wavelength closer to the trap-state PL peak position (Supplementary Fig. 21). Therefore, the positive TA peaks at 488 and 520 nm belong to the excited-state absorption (ESA) signals of trap state in AuAg-D NCs. We then focus on the femtosecond-TA spectra of AuAg-S NCs with single native AuAg NCs emitting state. Accordingly, the negative and positive TA bands centering at 420 and 475 nm are assigned to the ground-state bleaching (GSB) and ESA signals of AuAg NCs, respectively (Fig. 2c right). The very broad ESA band indicates the dense excited states of AuAg-S NCs[59]. Given the absolute dominance of native AuAg NCs emitting state over the trap-emitting state (93.2% vs 6.8% in terms of PL area), we attribute the TA signals of AuAg-H NCs with reference to that of AuAg-S NCs. Therefore, the positive TA band centering at 480 nm is classified as the ESA signal of native AuAg NCs emitting state in AuAg-H NCs (Fig. 2c middle). Thereafter, we carried

out global fittings of three TA maps to extract the time constants in their decay processes. As shown in Fig. 2d, e, the fits of 488 and 520 nm ESA kinetics of AuAg-D NCs give two-time components: 29.1 ns and >1 ns, corresponding to the electron injection process and radiative relaxation of excited electrons in trap state, respectively. In addition, the fits of 480 and 490 nm ESA kinetics of AuAg-H NCs give three decay components: 6.8 ps, 81.3 ps, and > 1 ns, corresponding to the structural relaxation (SR) in the sublevels of $S_1$, electron transfer from native AuAg NCs emitting state to the trap-emitting state, and radiative relaxation of excited electrons in $S_1$ of AuAg NCs, respectively. Finally, the fits of 475 and 570 nm ESA kinetics of AuAg-S NCs also give three decay components: 0.9 ps, 8.5 ps, and > 1 ns, corresponding to internal conversion (IC) of hot electrons from $S_n$ to the $S_1$ state ($S_n \rightarrow S_1$) in AuAg NCs, the SR process, and radiative relaxation process, respectively. Intriguingly, the initial absorbed H₂O molecules in AuAg-H NCs can

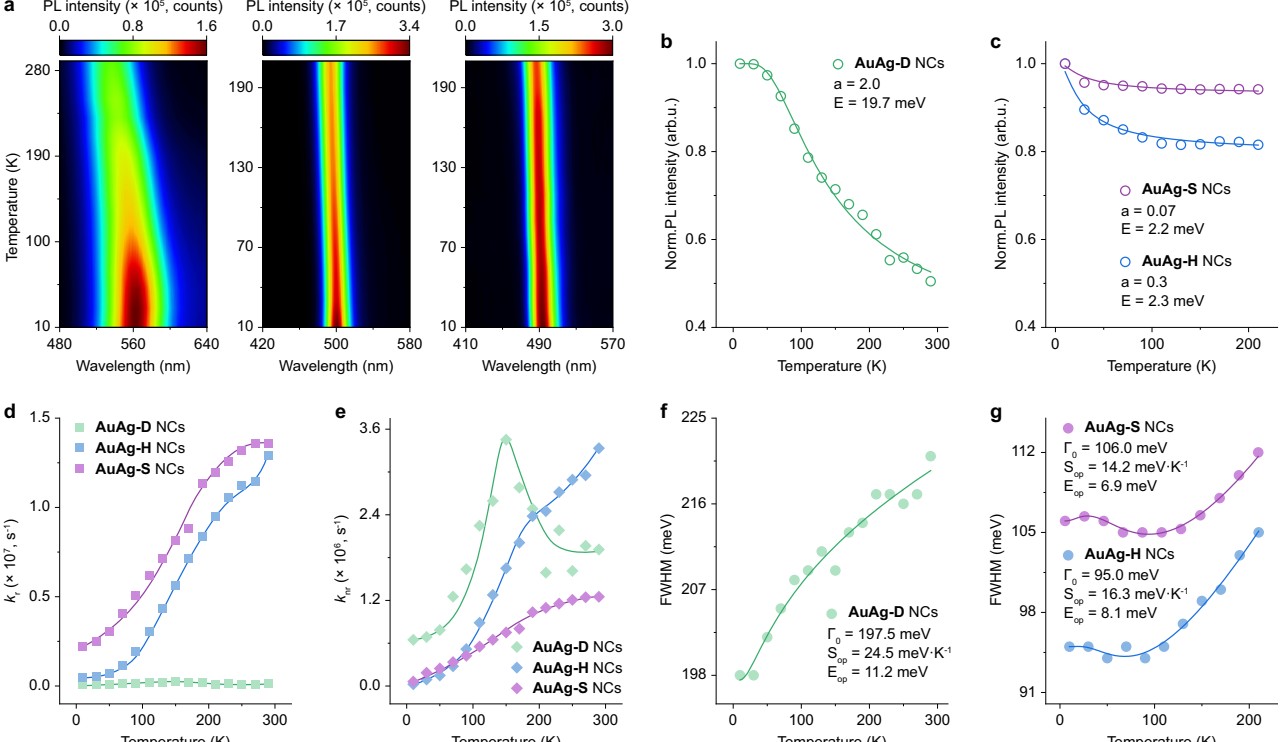

**Fig. 3 | Temperature-dependent optical properties of AuAg-D, AuAg-H, and AuAg-S NCs. a** Temperature-dependent PL maps of AuAg-D, AuAg-H, and AuAg-S NCs (from left to right). **b, c** Normalized integrated PL band intensities and corresponding fittings of AuAg-D, AuAg-H, and AuAg-S NCs as a function of temperature by using the Arrhenius equation. **d, e** Temperature-dependent $k_r$ and $k_{nr}$ of AuAg-D, AuAg-H, and AuAg-S NCs. **f, g** The excitonic linewidth and corresponding fittings of AuAg-D, AuAg-H, and AuAg-S NCs as a function of temperature by applying strong and weak electron-phonon coupling models, respectively. Source data are provided as a Source Data file.

significantly slow down the electron transfer process (from 29.1 to 81.3 ps) and further change the dominant radiative relaxation from the trap-state emission to the native AuAg NCs-state emission. The continued absorbed $H_2O$ molecules in AuAg-S NCs enable the electron transfer process to be completely prohibited (from 81.3 ps to undetected) while suppressing the non-radiative SR (from 6.8 to 8.5 ps) to final boost the emission of AuAg NCs.

Based on the transient absorption and PL experiments, the mechanism of tailoring trap-state emission with $H_2O$ molecules in AuAg NCs can be clarified. As shown in the proposed diagram in Fig. 2f, with 365 nm excitation, the electrons in AuAg NCs are directly excited to the $S_n$ state and then relax to the $S_1$ state through the 0.9 ps-IC process. Then, the excited electrons experience an SR process within 6.8–8.5 ps to relax to the lowest energy level of the $S_1$ state. Subsequently, the excited electrons either directly radiative relaxation from the $S_1$ state of AuAg NCs to the ground state of $S_0$ ($S_1 \rightarrow S_0$) to emit 480 nm sky-blue light with 41.6 ns lifetime, or undergo electron transfer to the trap-emitting state ($S_{trap}$) and then emit 536 nm green light to relax to the $S_0$ state ($S_{trap} \rightarrow S_0$) within ~1.0 µs. The significant effect of $H_2O$ molecules on the PL properties of AuAg NCs lies in their capability to modulate the electron transfer process and thus achieve a "directional shunt" of excited electrons at the $S_1$ energy level. As a result, the absence of $H_2O$ molecules in AuAg-D NCs would promote the electron transfer process (29.1 ps), and result in trap state-dominated emission with only 5.3% PLQY. A few $H_2O$ molecules in AuAg-H NCs would significantly decelerate the electron transfer process (81.3 ps) and switch the native $S_1$ state of AuAg NCs to the predominantly emissive state while increasing the PLQY to 79.5%. Finally, the sufficient $H_2O$ molecules in the aqueous solution of AuAg-S NCs will cut off the electron transfer process and reduce the non-radiative SR, thus further boosting the PLQY, up to 91.6%.

## PL origin and vibrational property

To better understand the PL origins and vibrational properties of AuAg-D, AuAg-H, and AuAg-S NCs, their corresponding temperature-dependent PL spectra were measured. Upon decreasing the temperature from 290 to 10 K, the PL profile of AuAg-D NCs experiences a continued red shift from 536 to 563 nm and becomes much sharper, suggesting a strong electron-phonon interaction in these NCs. In contrast, the PL profiles of AuAg-H and AuAg-S NCs show less redshift and change in peak shape at low temperatures, and therefore weaker electron-phonon interaction (especially for AuAg-S NCs with more absorbed $H_2O$ molecules, Fig. 3a and Supplementary Fig. 22). The integrated PL intensities of AuAg-D, AuAg-H, and AuAg-S NCs are found to increase by 2.0, 1.2, and 1.1 times from 290 to 10 K, respectively, which indicates that the PLQYs of AuAg-D, AuAg-H, and AuAg-S NCs reach up to 10.5%, 94.8%, and 97.3%, respectively. Furthermore, we plotted the quantitative temperature-dependent PL intensity evolutions of AuAg-D, AuAg-H, and AuAg-S NCs in Fig. 3b and c, respectively, where the PL intensities of all NCs at 10 K were set as unity. To extract the information on the structural origins of PL and thermally activated nonradiative relaxation in three NCs, these plots are then subjected to fit with the Arrhenius equation[60]:

$$I(T) = \frac{I_0}{1 + ae^{-E/k_B T}} \tag{2}$$

where $I_0$ and $I(T)$ refer to the integrated PL intensity at 10 K and a certain temperature of $T$, respectively. $a$ is the ratio of nonradiative and radiative probabilities, and $E$ is the activation energy for thermal-induced PL quenching. The corresponding fits give activation energies of phonon modes that are coupled with the PL of AuAg-D, AuAg-H, and AuAg-S NCs to be 19.7, 2.3, and 2.2 meV, respectively. Previous

**Table 1 | Fitting results of the FWHM of AuAg-D, AuAg-H, and AuAg-S NCs as a function of temperature**

| NCs | $\Gamma_0$ (meV) | $S_{op}$ (meV·K⁻¹) | $E_{op}$ (meV) | $S_{H_2O}$ (meV·K⁻¹) | $E_{H_2O}$ (meV) |
|---|---|---|---|---|---|
| AuAg-D | 197.5 | 24.5 | 11.2 | – | – |
| AuAg-H | 95.0 | 16.3 | 8.1 | 31.3 | 9.4 |
| AuAg-S | 106.0 | 14.2 | 6.9 | 46.0 | 11.1 |

theoretical and ultrafast experimental studies have demonstrated that the low-frequency phonon modes (<150 cm⁻¹) are typically caused by the Au-Au bond vibration in the kernel, while the relatively high-frequency phonon modes (<300 cm⁻¹) are attributed to the Au-S bond vibration in the staple motif in Au NCs[61]. Accordingly, we ascribe both the PLs coupled with 2.3 (18.4 cm⁻¹) and 2.2 meV (17.6 cm⁻¹) in AuAg-H and AuAg-S NCs to the vibrational (breathing or extensional) mode of the metal core (i.e., native emitting state) in AuAg NCs. However, the phonon mode of 19.7 meV (157.6 cm⁻¹) is much higher. Based on the dominated trap-state emission nature in AuAg-D NCs, this vibrational mode is better assigned to the Au atoms with exposed surfaces in the metal core because these sites in metal NCs are reported to have higher reactivity which makes them more accessible for $H_2O$ molecules. In addition, the $a$ value decreased from 2.0 (in AuAg-D NCs) to 0.3 (in AuAg-H NCs) and 0.07 (in AuAg-S NCs), suggesting that the staple motif vibration-induced nonradiative decay (SR process) has been dramatically suppressed by $H_2O$ molecules.

The temperature-dependent PL and lifetimes are combined to calculate the temperature-dependent $k_r$ and $k_{nr}$ of three AuAg NCs (Supplementary Fig. 23). As shown in Fig. 3d, the $k_r$ of AuAg-H and AuAg-S NCs show an obvious decreasing trend with the decline of temperature, manifesting that activation energy is present for radiative recombination of the native emitting state of AuAg NCs. In contrast, the $k_r$ of AuAg-D NCs keeps relatively stable at $10^{-6}$ magnitude regardless of the temperature change, which is a typical feature of trap-state emitting materials with temperature-insensitive excitons trapping process. Moreover, the $k_{nr}$ of AuAg-H and AuAg-S NCs exhibit a monotonic decreasing trend as the temperature decreases, revealing the suppression of the phonon population at low temperatures (Fig. 3e). A two-stage variation of the $k_{nr}$ of AuAg-D NCs is also informed from Fig. 3e where its $k_{nr}$ shows a monotonic decreasing dependence with the temperature decreases from 150 to 10 K, and $k_{nr}$ become irrelevant to temperature in the range of 290–150 K. This unique trend suggests that the $k_{nr}$ of AuAg-D NCs is dominated by the native emitting state of AuAg NCs at low temperatures and by the trap-emitting state at high temperatures, respectively. Overall, the poor PL properties in AuAg-D NCs are limited by the severely lagged radiative relaxation rate of the trap state, whereas the significantly boosted PLQY in AuAg-S NCs benefit from the reduced non-radiative relaxation rate, both of which are the result of modulation by $H_2O$ molecules.

To give deeper insights into the non-radiative electron-phonon coupling in three AuAg NCs, we investigated their PL broadening during temperature rise. The parameter of FWHM is employed to represent linewidth ($\Gamma$). The line widths of AuAg-D, AuAg-H, and AuAg-S NCs are extracted and plotted as a function of temperature. For AuAg-D NCs, a strong electron-phonon coupling model was applied to describe its linewidth broadening with the best fit (Fig. 3f)[57]:

$$\Gamma(T) = \Gamma_0 + \sqrt{S_{op}E_{op}\frac{1}{exp\left(\frac{E_{op}}{k_BT}\right) - 1}} \qquad (3)$$

where $S_{op}$ and $E_{op}$ refer to the coupling strengths and average energy of an electron with optical phonons (op), respectively. For AuAg-H and AuAg-S NCs, the corresponding best fits were found by using a weak

electron-phonon coupling model (Fig. 3g):

$$\Gamma(T) = \Gamma_0 + S_{op}\frac{1}{exp\left(\frac{E_{op}}{k_BT}\right) - 1} + S_{H_2O}exp\left(-\frac{E_{H_2O}}{k_BT}\right) \qquad (4)$$

Note that $S_{H_2O}$ and $E_{H_2O}$ were individually shown to describe the contribution of $H_2O$ molecules to their linewidth broadening. The fitting results are collected in Table 1. After excluding the intrinsic contribution of absorbed $H_2O$ molecules, we found the average energy $E_{op}$ of optical phonon modes gradually reduced from AuAg-D (11.2 meV) to AuAg-S NCs (6.9 meV), suggesting that the absorbed $H_2O$ molecules can greatly affect the periodical expansions/contractions of staple motifs in AuAg NCs. Meanwhile, the coupling strength of the optical phonon $S_{op}$ also dropped from 24.5 to 14.2 meV·K⁻¹. The changes in these parameters prove that the coupling of excited electrons with optical phonons in staple motifs mainly contributes to the non-radiative relaxation in AuAg NCs. This conclusion is also supported by the significantly declined $k_{nr}$ of three AuAg NCs at low temperatures due to the nature of the reduced population of high-energy optical phonons at low temperatures. The absorbed $H_2O$ molecules thus can suppress non-radiative relaxation induced by electron-optical phonon coupling to boost the optical properties of AuAg NCs.

## Structural insights into the defect passivation

For water-phase metal NCs, it is well-accepted that there are big challenges to acquiring high-quality single crystals for decoding their atomically precise structures. Based on this fact, we focus on investigating the local structures of AuAg NCs to clarify the defect type and corresponding passivation manner of $H_2O$ molecules. The positive matrix-assisted laser desorption ionization time-of-flight (MALDI-TOF) mass spectra of AuAg-S NCs show its composition to be $Au_8Ag_2(MPA)_6$ (Supplementary Fig. 24). Moreover, the high-resolution Au 4$f$ XPS spectra suggest that the Au $4f_{5/2}$ (88.1 eV) and $4f_{7/2}$ (84.3 eV) peaks of AuAg NCs are lying between that of Au(0) NPs (87.9 and 84.1 eV) and Au(I)-($p$-MBA) complexes (88.4 and 84.5 eV, Supplementary Fig. 25). Similarly, the high-resolution Ag 4$d$ XPS spectra suggest that the Au $4d_{3/2}$ (374.9 eV) and $4d_{5/2}$ (368.9 eV) peaks of AuAg NCs are lying between that of Ag(0) NPs (375.0 and 369.0 eV) and Ag(I)-($p$-MBA) complexes (374.7 and 368.7 eV, Supplementary Fig. 26). These results suggest that the as-synthesized AuAg NCs possess classical Au(0)Ag(0) @Au(I)Ag(I) core-shell structure. We further conducted peak deconvolutions of Au 4$f$ and Ag 4$d$ XPS spectra to quantify the amounts of Au(0), Au(I), and Ag(0), Ag(I) species in AuAg NCs. It was found that the ratio of Au(I):Au(0) is around 1:0.6 and that of Ag(I):Ag(0) is close to 1:1. Combined with the component of $Au_8Ag_2(MPA)_6$, we are informed that 5 Au(I) and 1 Ag(I) atoms bound to 6 MPA ligands to form staple motifs, and 3 Au(0) and 1 Ag(0) atoms collectively compose the metal core of AuAg NCs. Notably, the relative contents of Au(0) and Ag(0) in AuAg-H NCs are both slightly decreased compared to that in AuAg-D NCs, indicating that the absorbed $H_2O$ molecules may interact with both the Au(0) and Ag(0) atoms in the metal core and thereby influence their electron density.

The defect species in AuAg NCs was revealed by using electron paramagnetic resonance (EPR) measurements. As shown in Fig. 4a, the EPR spectrum of AuAg-D NCs gives a prominent EPR signal at $g = 2.003$. This unique EPR value can be attributed to oxygen vacancies ($O_v$) adjacent to metal atoms in AuAg NCs[62]. Intriguingly, the EPR signal intensity gradually weakened from AuAg-D to AuAg-H NCs and even almost disappeared in AuAg-S NCs, indicating that $H_2O$ molecules can efficiently passivate $O_v$. To clarify the interaction between $H_2O$ molecules and AuAg NCs, Raman spectra of dehydrated and hydrated AuAg NCs were collected. As shown in Fig. 4b, the Raman spectrum of AuAg-H NCs presents two additional Raman signals at 398 and 1403 cm⁻¹ compared to that of AuAg-D NCs, which

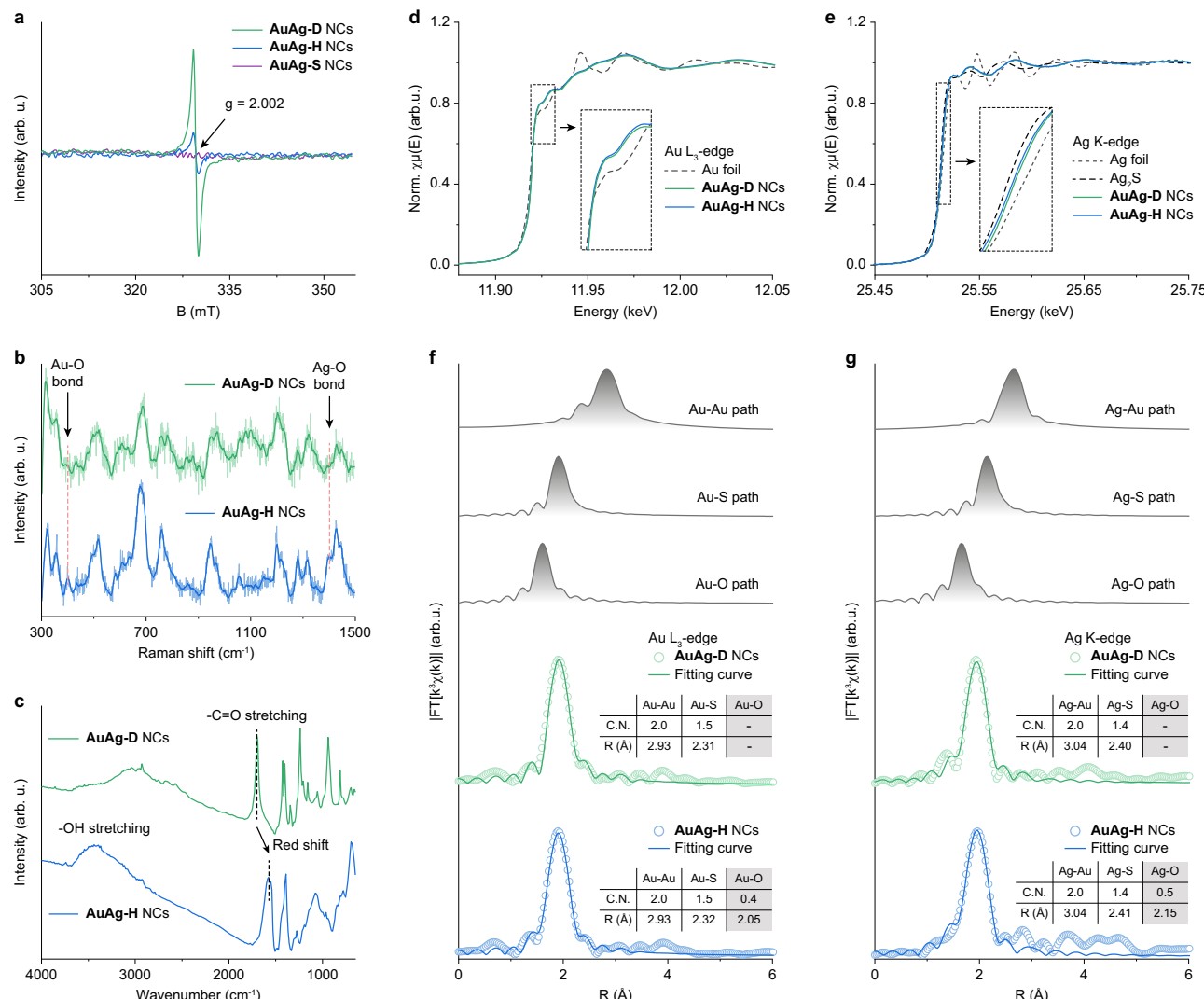

**Fig. 4 | Electronic properties and local structures of AuAg-D, AuAg-H, and AuAg-S NCs. a** EPR spectra of AuAg-D, AuAg-H, and AuAg-S NCs recorded at 90 K in liquid $N_2$. **b** Raman spectra of AuAg-D and AuAg-H NCs. The green and blue shadings are the experimental data, and the green and blue solid lines are the smoothed data of AuAg-D and AuAg-H NCs, respectively, which were calculated by using the Savitzky-Golay method with polynomial order of 2 and window points of 20. The Au-O and Ag-O bonds are highlighted by the red dotted lines. **c** SR-FTIR spectra of AuAg-D and AuAg-H NCs. The red shift of the signal of the carbonyl functional group is highlighted. **d** Au $L_3$-edge XANES spectra of Au foil, AuAg-D, and AuAg-H NCs. The inset shows the magnified XANES spectra in the energy range of 11.919–11.933 keV. **e** Ag K-edge XANES spectra of Ag foil, $Ag_2S$, AuAg-D, and AuAg-H NCs. The inset shows the magnified XANES spectra in the energy range of 25.510–25.520 keV. **f** Standard Au-Au, Au-S, and Au-O paths, and the experimental data and corresponding best fits of Au $L_3$-edge FT-EXAFS (shown in $k^3$-weighted *R*-space) of AuAg-D and AuAg-H NCs by using these paths. **g** Standard Ag-Au, Ag-S, and Ag-O paths, and the experimental data and corresponding best fits of Ag K-edge FT-EXAFS (shown in $k^3$-weighted *R*-space) of AuAg-D and AuAg-H NCs by using these paths. Source data are provided as a Source Data file.

are assigned to characteristic vibrational modes of Au-O and Ag-O bond[63,64], respectively. Therefore, the absorbed $H_2O$ molecules passivate $O_v$ in a manner of forming Au-O and Ag-O bondings. Synchrotron radiation-based Fourier transform infrared (SR-FTIR) spectra of AuAg-D and AuAg-H NCs were measured to investigate the interaction between $H_2O$ molecules and organic functional groups (Fig. 4c). The obvious bulge between 3600–3100 $cm^{-1}$ in the SR-FTIR spectrum of AuAg-H NCs is caused by the stretching vibration of -OH in absorbed $H_2O$ molecules. In addition, the characteristic stretching vibration of -C = O is found to redshift from 1703 (AuAg-D NCs) to 1575 $cm^{-1}$ (AuAg-H NCs), which contributes to the formation of hydrogen bondings between $H_2O$ molecules and the -C = O group in MPA ligands. This conclusion is also supported by the FTIR spectra of AuAg-D and AuAg-H NCs (Supplementary Fig. 27). The as-formed hydrogen bondings can stabilize MPA ligands and thus suppress nonradiative electron-optical phonon coupling in AuAg NCs.

Measuring the Au $L_3$- and Ag K-edge X-ray absorption fine structure (XAFS) spectroscopy of AuAg-D and AuAg-H NCs and fitting scattering paths that account for core and surface bonding revealed electronic and local structural changes that would be difficult to discern with other characterization techniques. Figure 4d and e display the Au $L_3$- and Ag K-edge X-ray absorption near-edge structure (XANES) spectra of AuAg-D and AuAg-H NCs, respectively. The intensity of the main indicator, the white line, reflects the relative amount of unoccupied 5 d and 4 d valence levels for Au and Ag in the NCs. Obviously, the Au $L_3$-edge white-line intensity of AuAg-H NCs is stronger than that of AuAg-D NCs (inset in Fig. 4d). Meanwhile, the Ag K-edge white-line intensity of AuAg-H NCs gets stronger than that of AuAg-D NCs and the absorption edge track goes away from the Ag foil and become closer to $Ag_2S$ (inset in Fig. 4e). The changes in the white-line intensities suggest that less electron density is localized in the 5 d and 4 d levels for Au and Ag atoms, which results in the increased

Au(Ag)(I) and decreased Au(Ag)(0) components after $H_2O$ absorption as demonstrated in their XPS analyses. Moreover, the varied near-edge resonances of Au and Ag indicate the $H_2O$ molecules-induced structural disorder. To further disclose this, Au $L_3$- and Ag K-edge Fourier-transformed extended XAFS (FT-EXAFS) measurements of AuAg-D and AuAg-H NCs with corresponding best fits in leveraging of Au(Ag)-Au, Au(Ag)-S, and Au(Ag)-O scattering paths were carried out and shown in Fig. 4f and g, respectively (Supplementary Figs. 28, 29 and Supplementary Tables 8, 9). Viewing from the end-on perspective, Au(Ag)-Au and Au(Ag)-S paths refer to Au(Ag)-Au interaction in the metal core, interfacial interaction between Au(I) atoms and protecting thiol (MPA) ligands in the staple motifs, respectively. However, the main difference is found for the Au(Ag)-O path. The absence of Au-O and Ag-O paths in the Au $L_3$- and Ag K-edge FT-EXAFS fits of AuAg-D NCs and the presence of these paths in that of AuAg-H NCs further confirm that the absorbed $H_2O$ molecules can generate both Au-O and Ag-O bonds to passivate $O_v$ and interact with AuAg NCs. The as-fitted coordination number (C.N.) and bond length (R) of the Au(Ag)-Au path in both AuAg-D and AuAg-H NCs are almost identical, indicating that the local structure of the metal core keeps stable. Nevertheless, although the as-fitted C.N. of Au(Ag)-S path is very close in AuAg-D and AuAg-H NCs, their corresponding R values undergo an obvious increase (from 2.31 to 2.32 Å for Au-S path and from 2.40 to 2.41 Å for Ag-S path) after $H_2O$ molecules absorption. These results prove that the above-mentioned structural disorder occurs mainly in the staple motif to adapt to the accommodation of $H_2O$ molecules in AuAg NCs. This structural disorder is also supported by the varied local structure of S atoms. As shown in the S K-edge XANES spectra (Supplementary Fig. 30a), the white-line intensity gets weakened for AuAg-H NCs than AuAg-D NCs, evidencing the redistribution of electron orbital density in S toward higher-valence states. The peak deconvolutions of high-resolution S 2p XPS spectra show the blue-shifted S $2p_{3/2}$ (from 161.40 to 161.45 eV) and $2p_{1/2}$ signals (from 162.50 to 162.83 eV, Supplementary Fig. 30b), confirming that the reduced electron density in S atoms due to the bonding of absorbed $H_2O$ molecules on AuAg NCs. Additional Zn K-edge XANES and FT-EXAFS analyses demonstrate that the introduced $H_2O$ molecules do not disturb the local structure of Zn atoms in AuAg NCs (Supplementary Figs. 31, 32 and Supplementary Table 11).

## Theoretical simulation and universality validation

We carried out theoretical simulations by using density functional theory (DFT) to confirm the above-mentioned passivation mechanism of $O_v$ by $H_2O$ molecules in AuAg NCs. To this end, a structural model for $Au_8Ag_2(MPA)_6$ is employed accordingly. Based on the above MALDI-TOF mass spectra and Au 4f, Ag 4d XPS results together with the previous theoretical studies on the structure of $Au_{10}(SR)_6$ NCs, the $Au_{10}(SR)_6$ model containing an $Au_6$ core and two $Au_2(SR)_3$ staple motifs is taken into consideration in this study[65]. The heteroatoms of Ag are then doped into the $Au_{10}(SR)_6$ model by replacing the pristine Au atoms. The most thermodynamically stable model of AuAg-D NCs with the lowest relative energies ($E_r$) is shown in Fig. 5a, which is composed of an $Au_5Ag$ core, an $Au_2(MPA)_3$ motif, and an $AuAg(MPA)_3$ motif (Supplementary Figs. 33, 34 and Supplementary Dataset 1). In addition, the theoretical average coordination numbers of Au-Au (C.N. = 2.0) and Ag-Au (C.N. = 2.0) bonds in this model are identical to the experimental values found in the Au $L_3$- and Ag K-edge FT-EXAFS fits, demonstrating the reliability of the used NCs model. The anchoring position of the absorbed $H_2O$ molecule in the $Au_8Ag_2(MPA)_6$ NCs is then optimized. It should clarify that four Au atoms in the $Au_5Ag$ core have already been bonded to the motif via Au-S interaction, while the remaining one Au and one Ag atom are bonded only to the metal. It is thereby rational to hypothesize that these Au and Ag atoms are more likely to accommodate $H_2O$ molecules. As expected, we found that the foreign $H_2O$ molecule preferentially interacts with the exposed Au atom in the core ($E_r = 0$ eV) in the

manner of forming an Au-O bond (denoted as AuAg-H-$H_2O$@Au NCs, Supplementary Dataset 2). In addition, the $H_2O$ molecule anchored to the Ag atom in the core in the form of an Ag-O bond is also feasible due to the negligible energetic barrier ($E_r = 0.02$ eV, denoted as AuAg-H-$H_2O$@Ag NCs, Fig. 5b, Supplementary Fig. 35 and Supplementary Dataset 3). These Au-O and Ag-O bondings would simultaneously disturb the electron density of Au and Ag atoms as verified by their corresponding XPS and FT-EXAFS results.

To further visualize the impact of $H_2O$ molecule-passivated $O_v$ on the electronic structure of AuAg NCs, the modeled absorption spectra for three NCs were calculated first. As shown in Fig. 5c, the simulated low-energy absorption peaks were found to be located at 410, 390, and 380 nm for AuAg-D, AuAg-H-$H_2O$@Au, and AuAg-H-$H_2O$@Ag NCs, respectively. An obvious blue shift of the absorption edge of AuAg NCs is manifested after anchoring the $H_2O$ molecule. All these results are closely matched with their experimental profiles. The calculated highest occupied molecular orbital (HOMO) and lowest unoccupied molecular orbital (LUMO) are composed of a contribution of more than 30% and 70% from Au atomic orbitals, respectively, whereas the contribution from Ag atomic orbitals is less than 10% (Fig. 5d–f and Supplementary Fig. 36). Therefore, the PL of AuAg NCs mainly comes from the radiative relaxation of excited electrons in Au atoms in the metal core. The HOMO and LUMO energetic positions of AuAg-D NCs are determined to be −5.97 and −3.83 eV, respectively, giving a theoretical bandgap energy of 2.14 eV. Especially, the contribution of O atomic orbitals to the LUMO is trivial (0.15%) due to the existence of $O_v$. For AuAg-H-$H_2O$@Au NCs, the anchoring of the $H_2O$ molecule on the Au atom would significantly shift its LUMO position to −3.50 eV, leading to the greatly enlarged $E_g$ value of 2.58 eV. Notably, the contribution of O states to LUMO increases to 5.25%. This result evidences that the absorbed $H_2O$ molecules might undergo orbital hybridization between the lone pairs of electrons in the O atoms and Au(Ag) d orbitals to form Au(Ag)-O bond directly without occurring H-O bond cleavage[66]. As a result, these $H_2O$ molecules can passivate $O_v$ defects and further tune the PL properties of AuAg NCs. However, the effect of the $H_2O$ molecule adsorbed on the Ag atom on the bandgap (2.23 eV) is not as pronounced as that on the Au atom, which may be related to the inherent low contribution of Ag atoms to the PL process.

To give more theoretical insights into the PL mechanism of three AuAg NCs, we carried out a hole-electron analysis to investigate their characteristics for electronic excitation transition, by using local excitation and charge transfer modes. According to the hole-electron theory, the dominant mode can be recognized by the S/D value, where S indicates the calculated overlap integral of hole-electron distribution, and D refers to the calculated distance between the hole and electron centroids[67]. The larger S associated with, the smaller D suggests the more evident local excitation. The computed S/D values are 0.602, 1.263, and 0.674 for AuAg-D, AuAg-H-$H_2O$@Au, and AuAg-H-$H_2O$@Ag NCs, respectively, indicating that the absorbed $H_2O$ molecules can promote the efficient local excitation. Charge density difference (CDD) between the ground and the excited states is calculated to visualize the electron transfer within three AuAg NCs (Supplementary Fig. 37). To reveal the subtle changes due to the absorbed $H_2O$ molecules, the distance of charge transfer ($D_{CT}$) based on the *electron density variation* during electron excitation has been calculated. The pristine AuAg-D NCs have a large $D_{CT}$ value of 1.092 Å. However, the incorporation of $H_2O$ molecules can remarkably shorten this value, especially for AuAg-H-$H_2O$@Au NCs, to 0.651 Å. The smaller $D_{CT}$ value means that the holes and electrons are easier to recombine to give efficient PL, and accordingly results in significantly improved PLQY.

We further synthesized a series of Au-D and AuCu-D NCs to validate the universality of the strategy of lighting up metal NCs by passivating trap states with $H_2O$ molecules. As shown in Fig. 5g, after fully hydrated in 56% RH air, both the resultant Au-H and AuCu-H NCs exhibit an obvious blue shift in PL peak (from 545 to 484 nm and from

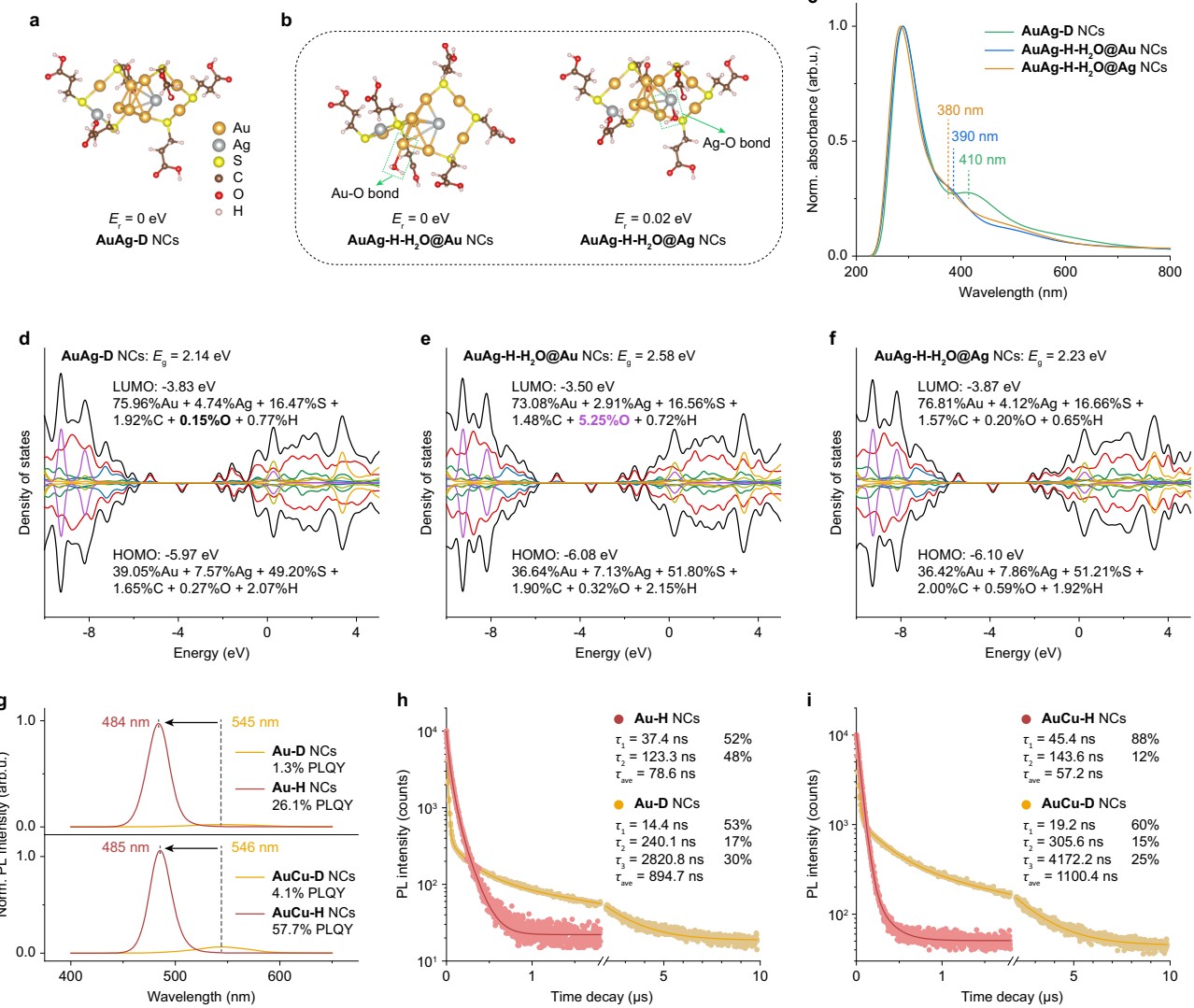

**Fig. 5 | Theoretical simulation and universality validation. a** The most stable model of AuAg-D NCs with the lowest relative energy ($E_r$ = 0 eV). **b** The most stable model of AuAg-H NCs including one $H_2O$ molecule anchoring to Au(0) atom ($E_r$ = 0 eV) or Ag(0) atom ($E_r$ = 0.02 eV) in the $Au_5Ag$ core. Two green dotted frames are marked to illustrate the as-formed Au-O and Ag-O bonds. **c** The simulated UV-vis absorption profiles of the optimized structural models. **d–f** Calculated densities of states of AuAg-D, AuAg-H·$H_2O$@Au, and AuAg-H·$H_2O$@Ag NCs, respectively. The black, red, green, cerulean-blue, yellow-green, purple, and orange solid lines refer to the total, Au, Ag, S, C, O, and H atomic orbital contributions to the HOMO and LUMO, respectively. **g** PL spectra of Au-D, Au-H (up plane) and AuCu-D, AuCu-H (bottom plane) NCs upon 365 nm excitation. **h** PL decays and corresponding fitting results of Au-D and Au-H NCs. **i** PL decays and corresponding fitting results of AuCu-D and AuCu-H NCs. All the samples are excited by a 370 nm pulsed laser. Source data are provided as a Source Data file.

546 to 485 nm, respectively) and significantly enhanced PL intensities. The corresponding absolute PLQYs are determined to increase from 1.3% (Au-D) to 26.1% (Au-H) and from 4.1% (AuCu-D) to 57.5% (AuCu-H), respectively (Supplementary Fig. 38), which is similar to the changes from AuAg-D to AuAg-H NCs. The corresponding PL decays of these NCs were collected upon 370 nm pulsed laser excitation. Both Au-D (14.4, 240.1, and 2820.8 ns) and AuCu-D NCs (19.2, 305.6, 4172.2 ns) show triexponential decay with an average PL lifetime of 894.7 and 1100.4 ns, respectively. Once they were hydrated, only two-time components can be fitted out for Au-H (37.4 and 123.3 ns) and AuCu-H NCs (45.4 and 143.6 ns), and the average PL lifetime significantly reduced to 78.6 and 57.2 ns, respectively (Fig. 5h, i). The disappearance of $\tau_3$ in both Au-H and AuCu-H NCs illustrates that the trap state has been passivated by the $H_2O$ molecules, and accordingly, the PL transforms from trap-state emission to native Au/AuCu NCs-state emission. As shown in the XANES spectra of Au-D, Au-H and AuCu-D, AuCu-H NCs, both the Au $L_3$- and Cu K-edge white-line intensity increased after hydration, indicating the reduced electron density in Au and Cu atoms (Supplementary Figs. 39, 40). The corresponding FT-EXAFS fits reveal that additional Au-O and Cu-O bonds have formed in Au-H and AuCu-H NCs to passivate $O_v$, compared to Au-D and AuCu-D NCs (Supplementary Figs. 41–43 and Supplementary Tables 8, 10). The Zn-K edge XANES and FT-EXAFS fits of these NCs demonstrate that the local structure around Zn atoms remains stable before and after hydration (Supplementary Figs. 44–47 and Supplementary Table 11).

## Discussion
In summary, we have demonstrated that the absorption of $H_2O$ molecules in AuAg NCs can shift the PL peak from 536 to 480 nm and significantly boost the absolute PLQY from 5.3% to 91.6%. The 536 nm green PL in AuAg-D NCs is proved to be the $O_v$-centered slow-radiation trap-state emission which originates from the electron transfer from the native AuAg NCs-emitting state. The absorbed $H_2O$ molecules in AuAg-H NCs can anchor to the Au and Ag atoms in the core by forming Au-O and Ag-O bonds. The passivation of inherent $O_v$ defects by $H_2O$ molecules would block the electron transfer channel and switch the

fast-radiation native AuAg NCs emission from the metal core to be the dominant PL pathway. The excess $H_2O$ molecules also benefit to stabilizing AuAg NCs to suppress the strength and energy of non-radiative electron-optical phonon coupling. As a result, the ultrahigh PLQY of 91.6% can be achieved in AuAg-S NCs. The applied strategy of lighting up metal NCs by passivating trap state with $H_2O$ molecules is proved to be universal to tuning the PL property of Au and AuCu NCs systems. This work is of significance not only because it achieves the high absolute PLQY of luminescent metal NCs, but also because it deepens the understanding of the trap-state emission in metal NCs and the significant role of solvent molecules in customizing their PL properties, which would be helpful to develop applicable metal NCs with tailorable emitting color and excellent PLQY.

## Methods

### Materials and reagents

Hydrogen tetrachloroaurate(III) trihydrate ($HAuCl_4 \cdot 3H_2O$, ≥ 49.0 Au basis) was purchased from Sigma Aldrich. Silver nitrate ($AgNO_3$, 99.99% metals basis), copper(II) chloride ($CuCl_2$, 99.99% metals basis), 3-mercaptopropionic acid (MPA, 99%), sodium hydroxide (NaOH, 96%), and zinc acetate (99.995% metals basis) were purchased from Aladdin Reagent Co. (Shanghai, China). Potassium bromide (99%) was purchased from Energy Chemical (Shanghai, China). All chemicals were used as received without additional purification. Ultrapure Millipore water (18.2 MΩ) was used throughout the experiments to dissolve regents and as reaction media.

### Synthesis of AuAg-S NCs

In a typical synthesis of AuAg-S NCs, 0.67 mL aqueous solution of $HAuCl_4$ (50 mM) and 0.13 mL aqueous solution of $AgNO_3$ (50 mM) were added to 9.2 mL of ultrapure water under 600 rpm stirring at room temperature. Then, 138 μL of MPA was added to the above aqueous solution. The body color of the mixture transformed from light yellow to white with the appearance of precipitation. After stirring for another 15 min, the pH of the aqueous solution was carefully tuned to 7.90 with the drops of NaOH solution (1 M). The white precipitation gradually dissolved, and the solution became colorless and transparent. To trigger the assembly of AuAg NCs, a 2 mL aqueous solution of $Zn(OAc)_2$ (0.1 M) was added. Under the illumination of 365 nm near-UV light, a change in the luminescent color of the solution from non-luminescent to intense orange-red can be observed by the naked eye. After an additional annealing time of 30 h at 18 °C, AuAg-S NCs with bright sky-blue emission can be acquired.

### Synthesis of AuAg-D NCs

After the reaction of AuAg-S NCs, a 2 mL aqueous solution of $Zn(OAc)_2$ (0.1 M) was added to the above aqueous solution of AuAg-S NCs. The mixtures were subjected to additional stirring for 5 min. The products were transferred into a 50 mL centrifuge tube and then subjected to centrifugation at $675 \times g$ for 5 min. The supernatant containing dissociative AuAg NCs was discarded. The AuAg NCs assembly was pre-solidified thoroughly by using liquid nitrogen and then freeze-dried at −65 °C and 5 Pa for 3 days to form the AuAg-D NCs. The resultant AuAg-D NCs powder was transferred to a glove box and sealed in a transparent scintillation vial for further usage.

### Synthesis of AuAg-H NCs

The AuAg-D NCs powder was aged in the ambient atmosphere (56% relative humidity, RH) to completely absorb the water molecules in the air. With the gradual absorption of water, a change in the luminescent color of AuAg NCs from green to sky-blue can be observed under 365 nm near-UV illumination. The AuAg-H NCs were acquired after aging for 30 min.

## Characterization

The scanning electron microscope (SEM) images were recorded on Requlus8100 (Hitachi) microscope operating at 2 kV. The attached energy dispersive X-ray detector (EDX) was employed to detect the elements' dispersion. Matrix-assisted laser desorption ionization time-of-flight (MALDI-TOF) mass spectra were tested on Brucker Autoflex speed TOF under a positive linear mode. Trans-2-[3-(4-tert-Butylphenyl)-2-methyl-2-propenyldidene] malononitrile (DCTB) was employed as the matrix for the mass measurements of all samples. Thermogravimetric analysis (TGA, ~ 3 mg sample used) was conducted in an $N_2$ atmosphere (flow rate ~ 50 mL/min) at a heating rate of 10 °C/min using a TGA550 analyzer (Waters). Brunauer-Emmet-Teller (BET) analysis was conducted on an automatic fast specific surface and porosity analyzer (Micromeritics ASAP 2460). The corresponding BET surface areas were calculated from the $N_2$ isotherms at 77 K (liquid nitrogen bath). Electron paramagnetic resonance (EPR) measurements were collected using an endor spectrometer (JEOL ES-ED3X) at 90 K in liquid nitrogen. X-ray photoelectron spectroscopy (XPS) spectra were collected on an ESCALAB250 spectrometer (Thermo Fisher). The C 1s peaks of all samples were calibrated to the reference signals for contaminated carbon (284.8 eV). UV-vis absorption spectra of the aqueous solution of AuAg-S NCs were recorded on a UV-1900i spectrometer (Shimadzu). Especially, solid-state UV-vis absorption spectra of AuAg-D and AuAg-H NCs powder were obtained by transforming their corresponding UV-vis diffuse reflectance spectra recorded on Lambda 1050 + ultraviolet-visible near-infrared spectrophotometer (PerkinElmer). All the PL and PL excitation (PLE) spectra and excitation-emission 2D maps were measured on an FLS1000 spectrofluorometer (Edinburgh Instruments). For the RH-dependent PL measurements of AuAg NCs, the RH condition around AuAg-D NCs in powder cuvette was controlled at certain values (2, 10, 15, 20, 30, 40, 50, and 56%, recorded by a humidity sensor) by regulating the flow rate of the air and the dry $N_2$ gas stream. The excitation source was fixed at 365 nm and the RH condition was maintained stable for 3 min before PL testing. For the temperature-dependent PL measurements of all metal NCs samples, AuAg-D and AuAg-H NCs powder were sealed in a liquid cuvette to prevent changes in PL properties due to the escape of water molecules (especially for AuAg-H NCs) in high vacuum conditions (typically at $10^{-6}$ mbar). An appropriate amount of glycerol was added to the aqueous solution of AuAg-S NCs to prevent the water from freezing at low temperatures. The cooling of samples was achieved by using a DE-202AI in situ variable temperature transmission system (Advanced Research Systems) which can control the temperature from 10 to 500 K with liquid helium cooling. The absolute PLQY of all AuAg NCs samples in aqueous solution and powder state were measured and calculated on the same instrument attached with an integrating sphere coating with a reflective $BaSO_4$ layer. For the detailed measurements of the PLQY of all samples, an Xe lamp with a fixed emission wavelength at 365 nm was used as the excitation source. Pure water and a $BaSO_4$ support were measured first in the integrating sphere and used as blank references for the samples in an aqueous solution and powder state, respectively. The calculation of absolute PLQY values was conducted on the built-in "Fluoracle" software (version 2.13.2). The values of PLQY were calculated by using the following equation: $PLQY = \frac{\int \lambda P(\lambda) d\lambda}{\int \lambda \{E(\lambda) - R(\lambda)\} d\lambda}$, where E(λ)/hv, R(λ)/hv, and P(λ)/hv are the number of photons in the spectrum of excitation, reflectance, and emission, respectively. Especially, for the measurements of near-bandgap excited PLQY, the excitation sources of 435, 410, and 410 nm were applied for AuAg-D, AuAg-H, and AuAg-S NCs, respectively. Time-resolved emission spectra (TRES) and temperature-dependent/room temperature PL lifetime measurements were carried out through the time-correlated single-photon counting (TCSPC) method on the same instrument using pulsed laser excitation sources. For the excitation power density-dependent PL intensity of all AuAg NCs samples, the

corresponding PL spectra were obtained by using an Olympus microscope and a Spectrapro 2300i spectrometer (Acton). The samples were excited by a 350 nm laser pulse and an external power attenuator was used to adjust the pump's power density. A spectrum from 400 to 650 nm was collected and the PL intensity was determined by the area integral of the spectrum. Fourier transform infrared (FTIR) spectra were recorded on a Nicolet Is50 spectrometer (Thermo Fisher) using a single attenuated total reflectance accessory. The scanning range is from 400 to 4000 cm$^{-1}$. Raman spectra were recorded on a high-resolution laser Raman spectrometer (HORIBA Jobin Yvon) attached with a 2017 Argon ion gas laser. A 532 nm laser was employed as an excitation source and the scanning range of Raman spectra was set as 300–1500 cm$^{-1}$.

## N$_2$ and O$_2$ saturated PL measurements

The AuAg-D NCs powder was preloaded in a powder cuvette first and vacuumed for 5 min with a vacuum pump to remove air. Then, it was purged with N$_2$/O$_2$ for 10 min, and the corresponding PL spectra were obtained upon 365 nm excitation, respectively. The N$_2$ and O$_2$ gases are ultrahigh-purity grade. The near-infrared (NIR) PL spectra of N$_2$/O$_2$-saturated AuAg-D NCs powders were recorded on a FLS1000 spectrofluorometer (Edinburgh Instruments) equipped with a 450 W continuous ozone-free xenon arc lamp.

## In situ high-pressure PL measurements

High-pressure experiments were performed using a symmetric diamond anvil cell (DAC) The sample was loaded into a hole (150 μm in diameter) of the T301 steel gasket, which was preindented to a thickness of 45 μm. The ruby fluorescence technique was adopted for the pressure calibration, and silicon oil (Aldrich, 150 mPa·s) was applied as a pressure transmitting medium (PTM) around the sample. A semiconductor laser with an excitation wavelength of 355 nm was employed for the high-pressure PL measurements of AuAg-D NCs. High-pressure PL spectra of AuAg-D NCs were recorded with an optical fiber spectrometer (Ocean Optics, QE65000). PL micrographs of the sample were obtained using a camera (Canon Eos 5D mark II) equipped with a microscope (Ecilipse TI-U, Nikon). The camera can record the photographs under the same conditions including exposure time and intensity.

## Femtosecond-transient absorption (TA) measurements

Femtosecond-TA spectroscopy was performed on a commercial Ti: Sapphire laser system (Spitfire Spectra-Physics). The laser pulse (~100 fs) in the ultraviolet and near-infrared wavelength was first generated in a 3.5 mJ regenerative amplifier system (Spitfire, Spectra-Physics) and optical parametric amplifier (OPA, TOPAS). A small portion of the laser fundamental was focused into a sapphire plate to produce a supercontinuum in the visible range, which overlapped in time and space with the pump. An electronically delayed supercontinuum light source with a sub-nanosecond pulse duration (EOS, Ultrafast Systems) was used as the probe. Multiwavelength transient spectra were recorded using dual spectrometers (signal and reference) equipped with array detectors whose data rates exceeded the repetition rate of the laser (1 kHz). Solution samples of AuAg-S NCs loaded in 1 mm path-length cuvettes were excited by the tunable output of the OPA (pump). While for the powder samples of AuAg-D and AuAg-H NCs, they are pre-mixed with KBr powder and then pressed into near-transparent sheets under 30 MPa for 10 min using a YP-2 tablet press mechanism (Shanyue scientific instrument) to ensure successful acquisition of the absorption signal. For the electron dynamics measurements of all samples, a 365 nm laser was employed as the excitation source. For the data analysis of all pristine TA data, background subtraction was first implemented with 5 spectra. The spectra were cropped in a certain wavelength range to wipe out interferences from spurious signals. Chirp correction was then carried out to calibrate the time zero position of different wavelengths. After that, time zero

correction was conducted to calibrate the initial time position of the whole TA map.

## Streak camera measurements

A 400 nm femtosecond laser (repetition rate of 1 kHz, pulse width of 80 fs) equipped with a streak camera (C10910, Hamamatsu) was used to investigate the femtosecond-fluorescence properties of AuAg NCs. A BBO crystal was used to generate 400 nm output from the 800 nm laser of a regenerative amplifier (SPTF-100F-1K-ACE, Spectra-Physics). The TRPL decay profile can be fitted by a deconvolution monoexponential or biexponential decay function with an impulse response function of 180 ps.

## X-ray absorption fine structure (XAFS) measurements

XAFS data of AuAg-D and AuAg-H NCs powder samples were collected at the Au L$_3$-edge ($E = 11919$ eV), Ag K-edge ($E = 25514$ eV), Zn K-edge ($E = 9659$ eV) in transmission mode and Cu K-edge ($E = 8979$ eV), S K-edge ($E = 2472$ eV) in fluorescence mode using a Lytle detector at the BL14W1 beamline of Shanghai Synchrotron Radiation Facility (SSRF). The samples were ground and uniformly daubed on the special adhesive tape. The acquired Fourier transformation extended XAFS (EXAFS) data were processed according to the standard procedures using the ATHENA module of Demeter software packages. The EXAFS spectra were obtained by subtracting the post-edge background from the overall absorption and then normalizing with respect to the edge-jump step. Subsequently, the χ(k) data were Fourier transformed to real ($R$) space using a Hanning window (dk = 1.0 Å$^{-1}$) to separate the EXAFS contributions from different coordination shells. To obtain the quantitative structural parameters around central atoms, least-squares curve parameter fitting was performed using the ARTEMIS module of Demeter software packages. The following EXAFS equation was used:

$$\chi(k) = \sum_j \frac{N_j S_0^2 F_j(k)}{k R_j^2} \cdot \exp[-2k^2 \sigma_j^2] \cdot \exp\left[\frac{-2R_j}{\lambda(k)}\right] \cdot \sin[2kR_j + \phi_j(k)] \quad (5)$$

The theoretical scattering amplitudes, phase shifts, and the photoelectron mean free path for all paths were calculated. Where $S_0^2$ is the amplitude reduction factor, $F_j(k)$ is the effective curved-wave backscattering amplitude, $N_j$ is the number of neighbors in the j$^{th}$ atomic shell, $R_j$ is the distance between the X-ray absorbing central atom and the atoms in the j$^{th}$ atomic shell (back scatterer), $\lambda$ is the mean free path in Å, $\phi_j(k)$ is the phase shift (including the phase shift for each shell and the total central atom phase shift), $\sigma_j$ is the Debye-Waller parameter of the j$^{th}$ atomic shell (variation of distances around the average $R_j$). The functions $F_j(k)$, $\lambda$, and $\phi_j(k)$ were calculated with the ab initio code FEFF9. The additional details for EXAFS simulations are given below. Since k$^3$ weighting amplifies the high k-range of the spectra it is more sensitive to heavy atoms such as Ag. Ag atoms, especially those forming NCs, do not always occupy fixed crystallographic positions. They are, therefore, associated with relatively large static disorders making them difficult to detect. Fitting the EXAFS data with k$^3$ weighting allows more useful information on the Ag shells to be obtained and gives the most reliable results. Therefore, all fits were performed in the R space with a k-weight of 3 while phase correction was also applied in the first coordination shell to make the R-value close to the physical interatomic distance between the absorber and shell scatterer. The coordination numbers of model samples were fixed as the nominal values. While the $S_0^2$, internal atomic distances R, Debye-Waller factor $\sigma^2$, and the edge-energy shift Δ were allowed to run freely.

## Synchrotron radiation-based Fourier transform infrared (SR-FTIR) measurements

SR-FTIR microspectroscopy measurements were performed at the BL01B beamline of SSRF, which is equipped with Nicolet 6700 Fourier

transform infrared spectrometer, Continuum XL FTIR microscope, and $32 \times$ Schwarzschild objective. Each spectrum within the wavenumber region (4000-600 $cm^{-1}$) was collected with a resolution of 4 $cm^{-1}$ and 64 co-added scans. The background spectrum was collected from the blank area of the sample primarily.

## Computational details

Density-functional theory (DFT) calculations were performed using the Gaussian 16 software package to obtain the electronic properties of these clusters[68]. Specifically, the Perdew-Burke-Ernzerhof (PBE) was employed, along with the all-electron basis set 6–31 G* for H, O, and S, and the effective-core basis set LANL2DZ for Au and Ag. Additionally, the Polarizable Continuum Model (PCM) using the integral equation formalism variant (IEFPCM) was applied with radii and non-electrostatic terms from Truhlar and coworkers' SMD solvation model to calculate the energy in water solution. The structure of AuAg-D NCs ($Au_8Ag_2(MPA)_6$) was optimized using the DFT method implemented in Vienna ab initio Simulation Package (VASP5.4.4) code[69]. Different software packages were chosen to optimize AuAg-D, AuAg-H-$H_2O$@Au, AuAg-H-$H_2O$@Ag NCs and their isomers, and simulate their corresponding electronic properties based on the consideration of guaranteeing the accuracy of simulated models but reducing the calculation time at the same time. The exchange-correlation was treated with the PBE functional, and the ion-electron interactions were described by the PAW method[70]. The van der Waals (vdW) interactions were included using the empirical DFT-D3 method[71]. The Gama-grid-mesh-based Brillouin zone k-points are set as $1 \times 1 \times 1$ with the cutoff energy of 400 eV. The convergence criteria were set as 0.03 eV $A^{-1}$ and $10^{-5}$ eV in force and energy, respectively. The electron and hole distributions of the models were constructed by Multiwfn and Visual Molecular Dynamics (VMD)[72–74]. The hole-electron analysis module of Multiwfn has been widely used to do the electron excitation analysis[75].

**Decay rate calculation**. All the PL decay curves recorded by TCSPC were subjected to fitting to extract the PL lifetimes according to the following monoexponential, biexponential, and triexponential decay models:

$$I_{(t)} = I_0 + A_1 exp^{\left(\frac{-t}{\tau_1}\right)} \tag{6}$$

$$I_{(t)} = I_0 + A_1 exp^{\left(\frac{-t}{\tau_1}\right)} + A_2 exp^{\left(\frac{-t}{\tau_2}\right)} \tag{7}$$

$$I_{(t)} = I_0 + A_1 exp^{\left(\frac{-t}{\tau_1}\right)} + A_2 exp^{\left(\frac{-t}{\tau_2}\right)} + A_3 exp^{\left(\frac{-t}{\tau_3}\right)} \tag{8}$$

where $I_{(t)}$ and $I_0$ denote to the PL intensity measured at time $t$ and $O$, $A_1$, $A_2$, and $A_3$ refer to the constants, $\tau_1$, $\tau_2$, and $\tau_3$ are the corresponding decay components, respectively. The average PL lifetime ($\tau_{ave}$) and the percentages of $\tau_1$ ($P_{\tau1}$), $\tau_2$ ($P_{\tau2}$), and $\tau_3$ ($P_{\tau3}$) were calculated according to the following Eqs. (9), (10), (11), and (12) respectively:

$$\tau_{ave} = \frac{A_1\tau_1^2 + A_2\tau_2^2 + A_3\tau_3^2}{A_1\tau_1 + A_2\tau_2 + A_3\tau_3} \tag{9}$$

$$P_{\tau1} = \frac{A_1\tau_1}{A_1\tau_1 + A_2\tau_2 + A_3\tau_3} \tag{10}$$

$$P_{\tau2} = \frac{A_2\tau_2}{A_1\tau_1 + A_2\tau_2 + A_3\tau_3} \tag{11}$$

$$P_{\tau3} = \frac{A_3\tau_3}{A_1\tau_1 + A_2\tau_2 + A_3\tau_3} \tag{12}$$

The radiative and non-radiative lifetimes ($\tau_r$, $\tau_{nr}$) were calculated according to Eqs. (13) and (14):

$$\tau_r = \frac{\tau_{ave}}{QY} \tag{13}$$

$$\tau_{nr} = \frac{\tau_{ave}}{1 - QY} \tag{14}$$

After that, the parameters of $k_r$ and $k_{nr}$ were calculated according to Eqs. (15) and (16):

$$k_r = \frac{1}{\tau_r} \tag{15}$$

$$k_{nr} = \frac{1}{\tau_{nr}} \tag{16}$$

## Reporting summary

Further information on research design is available in the Nature Portfolio Reporting Summary linked to this article.

## Data availability

The data supporting the findings of this study are available within the paper, Supplementary Information, and Source Data files. Extra data are available from the author upon request. Source data are provided in this paper.

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

## Acknowledgements

We thank Prof. M.Z. from the University of Science and Technology of China (USTC) for his significant discussion about the femtosecond-TA analysis. This work was supported by the National Natural Science Foundation of China (NSFC) (T2325015 and U21A2068 to X.B., 61935009 to Y. Zha., 12174151 to Z.W., 12304448 to T.L.).

## Author contributions

Y.Zho., X.W., Z.H., and Y.W. contributed equally to this work. S.G., Q.T., Z.W., and X.B. conceptualized the idea and co-supervised this work. Y.Zho., X.W., T.L., W.D., and H.Z. performed the synthesis of metal NCs, optical spectra measurements, and serial femtosecond-TA experiments. Z.H. carried out the DFT simulation. Y.Zho. and Y.W. conducted XAS characterizations. F.J. carried out streak camera measurements. Y.S. took digital photographs and movies. Y. Zho., X.W., Z.H., Y.W., W.D., F.J., H.Z., Z.Z., Z.W., Y.Zha., and X.B. discussed the experimental data and commented on the original draft. Y.Zho. and Z.W. wrote the manuscript.

## Competing interests

The authors declare no competing interests.
