## [Transparent Peer Review file · Nature Communications]

Lighting up metal nanoclusters by the H₂O-dictated electron relaxation dynamics

Corresponding Author: Professor Zhennan Wu

Version 0:

Reviewer comments:

Reviewer #1

(Remarks to the Author)

This work presents a fascinating study on the role of water molecules in metal nanoclusters, particularly their ability to passivate structural oxygen vacancies to tailor the excited-state electron dynamics and, in turn, achieve unusual color tuning from 536 to 480 nm and significantly boosted luminescence efficiency from 5.3% to 91.6%. The findings are compelling and contribute significant new insights into the design and optimization of luminescent nanomaterials. The relevant experimental and theoretical evidences supporting these conclusions are exceptionally robust and solid, offering clear and convincing data that strongly reinforce the proposed photophysical mechanisms. The richness of the data is impressive, with comprehensive analysis that elegantly links the structural properties of the nanoclusters with their optical behavior. The logical coherence of the overall structure and the depth of the investigation make this study a highly valuable contribution to the diverse fields. Overall, I found this work is novel and interesting, and the presented manuscript meets the high standards of Nature Communications. There are several aspects of this manuscript that will benefit readers in a variety of research areas. Therefore, I recommend its publication in Nature Communications with minor revisions.

1. The authors should refine the ABSTRACT section, less than 200 words would be better.
2. In Figures 2c and e, the authors should clarify the meaning of the color scales and the unit in the corresponding figure caption.
3. The authors have tested the absolute PLQY values for AuAg-D, AuAg-H, and AuAg-S NCs samples under 365 nm near-UV and band-edge excitation. Can the authors explain why the PLQY values are relatively higher under band-edge excitation?
4. Figure 12b in the SI file, the NIR spectra of N₂- and O₂-saturated AuAg-D NCs seem to have been acquired in different data interval modes, please check it.
5. The colored atoms in Figures 5a and b are suggested to be defined in the graphs instead of their figure captions.

Reviewer #2

(Remarks to the Author)

Zhong et al report an intriguing effect of H₂O (from air) on the photoluminescence of 3-mercaptopropionic acid (MPA) protected, ~10-atom AuAg, Au, and AuCu cluster powders. A significant enhancement of PL (QY from a few% to near unity) is observed. This universal effect on aqueous clusters is quite surprising.

The PL enhancing mechanism is rationalized to be the passivation of the defect or trap state (oxygen vacancies, O_v) by H₂O adsorption, which blocks electron transfer from the S1 state to the trap state, accompanied by emitting color shifting from 536 nm-green (trap state emission) to 480 nm-blue (S1 emission). Other effects of H₂O include the suppression of staple vibrations and the weakening of e-ph coupling.

Overall, the significant PL enhancement by passivating trap states with H₂O is quite surprising, though some details of the mechanism remain. I recommend this work be published after suitable revisions.

1. In the paragraph above "Mechanism...", statement "assign τ_1 and τ_2 to the non-radiative and radiative relaxation..." might not be correct. In your case, both lifetimes are from the PL decay measurement, rather than from transient absorption, so both

must be radiative components.

2. p16, it would be helpful to indicate a literature source for the 398 and 1403 cm^{-1} of the respective Au-O and Ag-O bond vibrations, e.g. a computational or experimental paper.

3. Regarding the evidence for Au-O and Ag-O bond formation after H₂O adsorption onto the clusters: This seems to indicate H-O bond cleavage (at least one H-O bond in the H₂O(ad) molecule), otherwise O-Ag and O-Au cannot be fully formed. But given the fact that the water effect is reversible, it is less likely that H₂O would be dissociated on the AuAg to form Au-OH or Au-O species. Some more explanation may be helpful.

4. If the passivation of metal traps by H₂O involves O-Ag and O-Au formation, then what about using alcohol R-OH (e.g. MeOH or EtOH) to passivate the trap states? Would a similar PL enhancement be observed?

Reviewer #3

(Remarks to the Author)

Yuan and co-workers demonstrate an innovative method to tune the luminescent properties of AuAg NCs by passivating oxygen vacancies with H₂O molecules. This leads to significant improvements in color tuning and PLQY, offering a promising strategy for developing high-performance luminescent materials. The study provides deep insights into trap chemistry and electron dynamics for advancing the field of metal NCs, potentially leading to new applications and technologies. The work presents useful viewpoints and the results are reliable. The paper is well organized with publishable level of quality, thus I recommend its publication after considering the following questions:

The role of DFT calculations in this paper is not significant. The interaction modes between the AuAg core and H₂O molecules should be investigated through theoretical calculations. The authors also could give further insight into luminescence mechanism through DFT calculations, eg. using hole-electron analysis to investigate the characteristics for electronic excitation.

Version 1:

Reviewer comments:

Reviewer #2

(Remarks to the Author)

The R1 manu is overall improved and publishable. The questions were addressed, in particular it's good to know that alcohols show a similar effect, though less than the effect of water. While Q1 needs more understanding, it's ok with me to publish R1 as is.

Reviewer #3

(Remarks to the Author)

The authors have answered all my questions, and I suggest publication as is.

Replies to reviewers' comments and descriptions of revisions made

Comments by Reviewer #1:

This work presents a fascinating study on the role of water molecules in metal nanoclusters, particularly their ability to passivate structural oxygen vacancies to tailor the excited-state electron dynamics and, in turn, achieve unusual color tuning from 536 to 480 nm and significantly boosted luminescence efficiency from 5.3% to 91.6%. The findings are compelling and contribute significant new insights into the design and optimization of luminescent nanomaterials. The relevant experimental and theoretical evidences supporting these conclusions are exceptionally robust and solid, offering clear and convincing data that strongly reinforce the proposed photophysical mechanisms. The richness of the data is impressive, with comprehensive analysis that elegantly links the structural properties of the nanoclusters with their optical behavior. The logical coherence of the overall structure and the depth of the investigation make this study a highly valuable contribution to the diverse fields. Overall, I found this work is novel and interesting, and the presented manuscript meets the high standards of Nature Communications. There are several aspects of this manuscript that will benefit readers in a variety of research areas. Therefore, I recommend its publication in Nature Communications with minor revisions.

Reply: We sincerely appreciate your very positive comments on the novelty and significance of our study. We would also like to thank your inspiring and constructive comments and suggestions, which have been taken into careful consideration in this revision. Please see below for a point-to-point response to your specific comments/suggestions. We hope the quality of our revised manuscript has been greatly improved according to your comments/suggestions and can be considered for publication in *Nature Communications*.

1. *The authors should refine the ABSTRACT section, less than 200 words would be better.*

Reply: Thank you for this kind suggestion. We have refined the content of the Abstract section to ensure that it is less than 200 words.

Revisions:

Page 2, Line 23-38:

“The modulation of traps has found attractive attention to optimize the performance of luminescent materials, while the understanding of trap-involved photoluminescence (PL) management of metal nanoclusters (NCs) greatly lags behind, thus extensively impeding their increasing acceptance as the promising chromophores. Here, we report an efficient passivation of the structural oxygen

vacancies in AuAg NCs by leveraging the H₂O molecules, achieving a sensitive color tuning from 536 to 480 nm and remarkably boosting PL quantum yield (PLQY) from 5.3% (trap-state emission) to 91.6% (native-state emission). In detail, favored electron transfer relevant to the structural oxygen vacancies of AuAg NCs contributes to the weak trap-state emission, which is capable of being restrained by the H₂O molecules by taking Au-O and Ag-O bonds. This scenario allows the dominated native-state emission with a faster radiative rate. In parallel, the H₂O molecules can rigidify the landscape of AuAg NCs leveraging on the hydrogen bonding, thus enabling an efficient suppression of electron-optical phonon coupling with a decelerated non-radiative rate. The presented study deepens the understanding of tailoring the PL properties of metal NCs by manipulating surface trap chemistry and electron relaxation dynamics, which would shed new light on luminescent metal NCs with customizable performance.”

2. *In Figures 2c and e, the authors should clarify the meaning of the color scales and the unit in the corresponding figure caption.*

Reply: Thank you for this insightful comment. The scaling of the color scales in Figures 2c and 3 in the main text is the absorption intensity, and the unit is milli-optical density (mOD). We have added this clarification in the corresponding figure caption.

Revisions:

Page 11, Line 303-304:

“The scaling of the color scales is the absorption intensity, and its unit is milli-optical density (mOD).”

3. *The authors have tested the absolute PLQY values for AuAg-D, AuAg-H, and AuAg-S NCs samples under 365 nm near-UV and band-edge excitation. Can the authors explain why the PLQY values are relatively higher under band-edge excitation?*

Reply: Thank you for this insightful comment. The absolute PLQYs for **AuAg-D**, **AuAg-H**, and **AuAg-S** NCs were recorded to be 5.3%, 79.5%, and 91.6% upon near-bandgap energy excitation (435 nm for **AuAg-D** NCs, and 410 nm for **AuAg-H** and **AuAg-S** NCs), which are relatively higher than the values measured under 365 nm excitation (6.6%, 76.7%, and 85.1%, respectively). These results can be attributed to that bandgap energy excitation reduces the non-radiative relaxation process of excited-state electrons in AuAg NCs compared to 365 nm near-UV excitation. The bandgap energy excitation is closer to the intrinsic absorption edge of the AuAg NCs. It can more efficiently excite the electrons directly into the lowest energy level of the S₁ state, thus reducing the chance that the excited-state electrons return to the ground state through the non-radiative relaxation path of the higher energy levels.

4. *Figure 12b in the SI file, the NIR spectra of N₂- and O₂-saturated AuAg-D NCs seem*

to have been acquired in different data interval modes, please check it.

Reply: Thank you for this insightful comment. We have recollected the NIR emission spectra of N₂- and O₂-saturated **AuAg-D** NCs in the same data-interval mode (**Figure R1**). It was found that no characteristic ¹O₂ emission peak was detected at ~1275 nm for both N₂- and O₂-saturated **AuAg-D** NCs, suggesting that emitter II does not belong to triplet-state phosphorescence.

Figure R1. NIR PL spectra of N₂- and O₂-saturated **AuAg-D** NCs powders.

Revisions:

SI, Page 14, Figure S13b:

Figure R1 has been included as **Figure S12b** in the supporting information.

5. The colored atoms in Figures 5a and b are suggested to be defined in the graphs instead of their figure captions.

Reply: Thank you for this insightful comment. We have defined the colored atoms in Figure 5a in the main text (**Figure R2** in this response).

Figure R2. The most stable model of **AuAg-D** NCs with the lowest relative energy ($E_r = 0$ eV).

Revisions:

Page 20, Figure 5a:

Figure R2 has been included as **Figure 5a** in the main text.

Page 20, Figure 5a caption:

The statement “Color scheme: orange, gold; grey, silver; yellow, sulfur; brown, carbon; red, oxygen; pink, hydrogen” has been deleted in the figure caption of **Figure 5a** in the main text.

Comments by Reviewer #2:

Zhong et al report an intriguing effect of H₂O (from air) on the photoluminescence of 3-mercaptopropionic acid (MPA) protected, ~10-atom AuAg, Au, and AuCu cluster powders. A significant enhancement of PL (QY from a few% to near unity) is observed. This universal effect on aqueous clusters is quite surprising.

The PL enhancing mechanism is rationalized to be the passivation of the defect or trap state (oxygen vacancies, Ov) by H₂O adsorption, which blocks electron transfer from the S₁ state to the trap state, accompanied by emitting color shifting from 536 nm-green (trap state emission) to 480 nm-blue (S₁ emission). Other effects of H₂O include the suppression of staple vibrations and the weakening of e-ph coupling.

Overall, the significant PL enhancement by passivating trap states with H₂O is quite surprising, though some details of the mechanism remain. I recommend this work be published after suitable revisions.

Reply: We sincerely appreciate your very positive comments on the novelty and significance of our work. We would also like to thank you for your inspiring and constructive comments and suggestions, which have been carefully considered in this revision. Please see below for a point-to-point response to your specific comments/suggestions.

1. In the paragraph above "Mechanism...", statement "assign τ_1 and τ_2 to the non-radiative and radiative relaxation..." might not be correct. In your case, both lifetimes are from the PL decay measurement, rather than from transient absorption, so both must be radiative components.

Reply: Thank you for this insightful comment. First, we would like to clarify that lifetimes obtained from PL decay measurements can include non-radiative components.

i) PL lifetime is indeed a time scale that describes the PL process. However, this parameter is not only determined by the radiative relaxation (i.e., PL), but also affected by all the possible relaxation in the luminescent materials together, which includes the non-radiative relaxation (the excited-state electrons dissipate energy as heat or otherwise by interacting with lattices, phonons, defect states, or other electrons without emitting photons, *Nature*, **2009**, 459, 234; *Science*, **2015**, 350, 1065; *ACS Energy Lett.*, **2020**, 5, 1380). This is because the PL lifetime is defined as the average time required for an excited state electron to return from the excited state to the ground state, and this

return path can be either radiative or non-radiative.

ii) PL lifetime is a representation of the overall decay of excited state electrons. When a material is excited by light, the exponential decay of the number of excited-state electrons ($N(t)$) with time can be described as follows (*Phys. Rev. Lett.*, **2004**, 93, 260601):

$$N(t) = N_0 e^{-t/\tau}$$

where N_0 is the initial number of excited-state electrons, t is the time, and τ refers to the overall PL lifetime which represents the average time the electron stays in the excited state. However, both radiative and non-radiative relaxations can lead to the extinction of excited-state electrons, which in turn affects the PL lifetimes.

iii) The relationship between the overall PL lifetime (τ) and the radiative (τ_r) and non-radiative lifetimes (τ_{nr}) of the luminescent material can be described as follows (*Opt. Express*, **2014**, 22, 17959; *Phys. Status Solidi B*, **2024**, 261, 2400029):

$$\frac{1}{\tau} = \frac{1}{\tau_r} + \frac{1}{\tau_{nr}}$$

The total PL lifetime is remarkably reduced even when τ_{nr} is small. In our work, **AuAg-H** NCs have an overall τ value of 41.6 ns which is composed of two time components, namely 29.5 (τ_1 , 73%) and 72.8 ns (τ_2 , 27%). **AuAg-S** NCs exhibit a single time component of 69.5 ns. Therefore, the relatively reduced PL lifetime of **AuAg-H** NCs compared to that of **AuAg-S** NCs is caused by the non-radiative relaxation.

iv) Quantum yield (QY) of luminescent materials can be understood as the ratio of the radiative relaxation to the overall relaxation, and the relationship between QY and PL lifetime can be described as follows (*Chem. Rev.*, **2008**, 108, 1245):

$$QY = \frac{\tau/\tau_r}{1 + \tau/\tau_{nr}}$$

The presence of non-radiative relaxation in luminescent materials can lead to a reduced PLQY. In contrast, luminescent materials with near-unity PLQY typically exhibit a single PL lifetime component (i.e., τ_r). In our work, **AuAg-H** and **AuAg-S** NCs have PLQY values of 79.5% and 91.6%, respectively. The relatively lower PLQY in **AuAg-H** NCs is ascribed to the additional non-radiative relaxation component.

Second, we would like to clarify the assignments of τ_1 and τ_2 in **AuAg-D** and **AuAg-H** NCs to the non-radiative and radiative relaxation, respectively. It should be noted that the room-temperature PL lifetimes of all NCs were calculated by measuring their time-resolved emission spectra (TRES) through the time-correlated single-photon counting (TCSPC) method. However, such measurements can only record the overall change in the number of electrons in the excited state, without being able to directly distinguish between specific relaxation channels. Regarding this, we have carried out the temperature-dependent PL lifetime measurements in the temperature range of 10-290 K, based on the principle that non-radiative relaxation (e.g., electron-phonon coupling) would be significantly weakened, while radiative relaxation can be increased at low temperatures (*Adv. Mater.*, **2010**, 22, 1689; *J. Am. Chem. Soc.*, **2023**, 145, 19969). As shown in **Figure R3ac**, the proportion of τ_1 decreased from 57.5% to 19.2% while that

of τ_2 increased from 16.5% to 56.6% in **AuAg-D** NCs from 290 to 30 K. Similarly, the proportion of τ_1 decreased from 73.4% to 41.9% while that of τ_2 increased from 26.6% to 58.1% in **AuAg-H** NCs from 290 to 10 K (**Figure R3bd**). It is thereby rational to assign τ_1 and τ_2 to the non-radiative and radiative relaxation of excited-state electrons in AuAg NCs, respectively.

Figure R3. Temperature-dependent PL lifetime. **a, b** PL lifetimes of **AuAg-D** and **AuAg-H** NCs collected from 10 to 290 K with a temperature interval of 20 K. A 370 nm pulsed laser was used as the excitation source and the monitoring wavelength was set at 536 and 482 nm for **AuAg-D** and **AuAg-H** NCs samples, respectively. **c, d** Variations of τ_1 , τ_2 , and τ_3 in the PL decays of **AuAg-D** NCs and τ_1 , τ_2 in the PL decays of **AuAg-H** NCs with the decrease of temperature.

2. *p16*, it would be helpful to indicate a literature source for the 398 and 1403 cm^{-1} of the respective Au-O and Ag-O bond vibrations, e.g. a computational or experimental paper.

Reply: Thank you for this kind suggestion. We have added related literatures to support the assignments of Au-O and Ag-O bond vibrations in the Raman spectra.

Revisions:

Page 16, Line 444-445:

“...which are assigned to characteristic vibrational modes of Au-O and Ag-O bond^{63,64}, respectively.”

Page 35, Line 1023-1028:

Additional references have been added to support our statement:

63. Wang, L., Uosaki, K. & Noguchi, H. Effect of electrolyte concentration on the solvation structure of gold/LITFSI–DMSO solution interface. *J. Phys. Chem. C* 124, 12381–12389 (2020).

64. Wang, C.-B., Deo, G. & Wachs, I. E. Interaction of polycrystalline silver with oxygen, water, carbon dioxide, ethylene, and methanol: In situ raman and catalytic studies. *J. Phys. Chem. B* 103, 5645–5656 (1999).

3. *Regarding the evidence for Au-O and Ag-O bond formation after H₂O adsorption onto the clusters: This seems to indicate H-O bond cleavage (at least one H-O bond in the H₂O(ad) molecule), otherwise O-Ag and O-Au cannot be fully formed. But given the fact that the water effect is reversible, it is less likely that H₂O would be dissociated on the AuAg to form Au-OH or Au-O species. Some more explanation may be helpful.*

Reply: Thank you for this insightful comment. We are sorry that we didn't articulate this issue well in our first submission. We totally agree with your viewpoint that the formation of Au-O and Ag-O bonds is attributed to the adsorption and desorption of H₂O molecules instead of H-O bond cleavage, which results in the reversible impact of water effect on the PL tuning. We would like to clarify this point in two aspects as follows:

i) It is more stable for H₂O molecules to form Au(Ag)-O bonds directly with Au(Ag) atoms than to form these bonds after H-O bond cleavage. We have theoretically calculated the free energies of Au(Ag)-O bonds in **AuAg-H-H₂O@Au** and **AuAg-H-H₂O@Ag** NCs in the stable state where H₂O molecules are directly adsorbed on AuAg clusters (denoted as *H₂O), in the transition state where H-O bond cleavage in this H₂O molecule occurs (denoted as TS), and in the stable state after H-O bond cleavage (denoted as *OH*H, where the generated hydroxyl (*OH) interacts with Au(Ag) to form Au(Ag)-O bonds). As shown in **Figure R4**, the cleavage of the H-O bond in the adsorbed H₂O molecule requires overcoming a large energy barrier, namely 2.49 and 1.62 eV for **AuAg-H-H₂O@Au** and **AuAg-H-H₂O@Ag** NCs, respectively. The final free energies for H-O bond cleavage and to form stable Au(Ag)-O bonds in **AuAg-H-H₂O@Au** and **AuAg-H-H₂O@Ag** NCs were determined to be 1.75 and 1.26 eV, respectively. All the free energies in TS and *OH*H states are much larger than that of pristine *H₂O state, which means that the Au(Ag)-O bonds are formed between AuAg NCs and the intact H₂O molecules instead of *OH after H-O bond cleavage of H₂O molecules.

ii) The fundamentals of the formation of Au(Ag)-O bonds between Au(Ag) atoms and H₂O molecules. In our work, we have theoretically calculated the contributions of atomic orbitals (AOs) to the HOMO and LUMO of **AuAg-D** and **AuAg-H-H₂O@Au** NCs (**Figure R5**). A huge difference can be found in the components of LUMO, where the contribution of O increases from 0.15% to 5.25%. This result supports that the lone pairs of O in the adsorbed H₂O molecules have undergone orbital hybridization with the *d* AOs of the metal in the AuAg NCs through the formation of the Au-O bond. Similar conclusions were also reported by other researchers (*Phys. Rev. B*, **2004**, 69,

205411; *J. Phys. Chem. A*, **2008**, 112, 1313; *J. Chem. Phys.*, **2012**, 137, 114709; *J. Comput. Theor. Nanosci.*, **2014**, 11, 511). We have added more explanation in the main text to state this point.

Figure R4. Calculated free energies of AuAg-H-H₂O@Au and AuAg-H-H₂O@Ag NCs with different reaction coordinates. Note that *H₂O stands for H₂O molecule adsorbed on the surface of AuAg NCs, TS is the transition state of H₂O molecule during H-O bond cleavage, and *OH*H refers to the state of H₂O molecules after H-O bond cleavage.

Figure R5. Calculated densities of states of (a) AuAg-D and (b) AuAg-H-H₂O@Au NCs. Insets show the detailed contributions of atomic orbitals to the HOMO and LUMO.

Revisions:

Page 21, Line 566-570:

“This result evidences that the adsorbed H₂O molecules might undergo orbital hybridization between the lone pairs of electrons in the O atoms and Au(Ag) *d* orbitals to form Au(Ag)-O bond directly without occurring H-O bond cleavage⁶⁶. As a result, these H₂O molecules can passivate O_v defects and further tune the PL properties of AuAg NCs.”

Page 35, Line 1031-1033:

Additional reference has been added to support our statement:

66. Ranea, V. A., Michaelides, A., Ramírez, R., Vergés, J. A., Andres, P. L. & King, D. A. Density functional theory study of the interaction of monomeric water with the Ag{111} surface. *Phys. Rev. B* 64, 205411 (2004).

4. If the passivation of metal traps by H₂O involves O-Ag and O-Au formation, then what about using alcohol R-OH (e.g. MeOH or EtOH) to passivate the trap states? Would a similar PL enhancement be observed?

Reply: Thank you for this insightful comment. To demonstrate the passivation mechanism of structural traps involving the formation of Au-O and Ag-O bonds, we have measured the PL properties of **AuAg-D** NCs added with H₂O and different alcohols as solvents. As shown in **Figure R6a**, **AuAg-D** NCs added with H₂O, methanol, and ethylene glycol show sky-blue emission, where the emission intensity of the sample with the addition of H₂O was the brightest. On the contrary, **AuAg-D** NCs added with ethanol, isopropanol, n-propanol, and 1,2-propylene glycol solvents exhibit bleak green emission which is close to the emission of raw **AuAg-D** NCs. We further measured the PL spectra of corresponding samples (**Figure R6b**). The peak positions were found to be located at 481, 483, 490, 515, 513, 513, and 515 nm for **AuAg-D** NCs added with H₂O, methanol, ethylene glycol, ethanol, isopropanol, n-propanol, and 1,2-propylene glycol solvents, respectively, which agrees with the observed emitting color in **Figure R6a**. In addition, the emission intensities of all sky-blue-emitting samples were stronger than that of green-emitting samples, indicating that alcohol solvents can passivate structural traps in **AuAg-D** NCs but to varying degrees. The discrepancies in the passivation effect of these alcohol solvents can be attributed to their inherent differences in the solvent polarity and steric effect. i) The solvent polarity effect. H₂O molecules have a relatively stronger polarity ($\epsilon = 78.5$) which means that the lone pairs of electrons on its O atoms can form a strong coordination with the *d* orbitals of Au(Ag) atoms, thus filling and passivating the oxygen vacancy defects efficiently. On the contrary, other alcohol solvents have weak polarities due to that the electron density distribution of their hydroxyl group (-OH) is affected by the steric effect and electron push-pull effect of adjacent alkyl groups, which results in a weak passivation of structural traps. However, alcohol solvents with higher polarity were found to enable **AuAg-D** NCs to show a more pronounced blue shift and higher emission intensity (e.g., ethylene glycol and methanol with the polarity of 37.7 and 32.6, respectively). These results support that the polarity of solvent molecules significantly affects their interactions with metal NCs, and further the physicochemical properties of metal NCs. Similar conclusions have been widely reported by other researchers (*Nano Lett.*, **2014**, 14, 2670; *Nanoscale*, **2014**, 6, 12626; *J. Mol. Graphics Modell.*, **2021**, 105, 107866; *Int. J. Nanopart.*, **2008**, 1, 212). ii) The steric effect. The alkyl groups of alcohol molecules may increase the steric effect on the **AuAg-D** NCs surface due to their larger size, making it more difficult to approach and further passivate structural traps. On the contrary, H₂O molecules were reported to be small enough to approach metal NCs and form interactions (*J. Am. Chem. Soc.*, **2018**, 140, 15430; *J. Am. Chem. Soc.*, **2011**, 133, 11632; *J. Phys. Chem. Lett.*, **2013**, 4, 2943; *Nanoscale*, **2014**, 6, 12626). We have supplemented some necessary discussions in the main text to state this point.

Figure R6. Universal validation with different alcohols. **a** Digital photos show **AuAg-D** NCs added with different alcohols under sunlight (top plane) and 365 nm near-UV light illumination (bottom plane). The permittivity (ϵ) of H₂O is 78.5. MeOH: methanol ($\epsilon = 32.6$); EG: ethylene glycol ($\epsilon = 37.7$); EtOH: ethanol ($\epsilon = 24.3$); IPA: isopropanol ($\epsilon = 19.9$); n-PrOH: n-propanol ($\epsilon = 20.1$); PG: 1,2-propylene glycol ($\epsilon = 32.0$). **b** PL spectra of raw **AuAg-D** NCs and **AuAg-D** NCs added with different alcohols. The excitation wavelength was set at 365 nm. The samples were prepared by injecting 5 mL alcohol solvent into translucent scintillation vials with ~ 3 mg **AuAg-D** NCs powder. All of the above operations are completed in a glove box in a N₂ atmosphere.

Revisions:

Page 5, Line 125-129:

“In addition, the addition of other alcohol solvents (i.e., methanol, ethylene glycol, ethanol, isopropanol, n-propanol, 1,2-propylene glycol) was found to improve the PL performance of **AuAg-D** NCs to varying degrees, but none of them were as pronounced as the addition of H₂O (Supplementary Fig. 5). These results might be related to the higher polarity and smaller steric effect of H₂O molecules.”

SI, Page 6, Figure S5:

Figure R6 has been included as **Figure S5** in the supporting information.

Comments by Reviewer #3:

Yuan and co-workers demonstrate an innovative method to tune the luminescent properties of AuAg NCs by passivating oxygen vacancies with H₂O molecules. This leads to significant improvements in color tuning and PLQY, offering a promising strategy for developing high-performance luminescent materials. The study provides deep insights into trap chemistry and electron dynamics for advancing the field of metal NCs, potentially leading to new applications and technologies. The work presents useful viewpoints and the results are reliable. The paper is well organized with publishable level of quality, thus I recommend its publication after considering the following questions.

Reply: We sincerely appreciate your very positive comments on the novelty and significance of our work. We would also like to thank your detailed review and constructive comments. All the concerns about DFT calculations in this work have been seriously considered and revised accordingly. Please see below for a point-to-point response to your specific comments/suggestions.

1. *The role of DFT calculations in this paper is not significant. The interaction modes between the AuAg core and H₂O molecules should be investigated through theoretical calculations.*

Reply: Thank you for this insightful comment. We have conducted theoretical calculations to investigate the interaction modes between AuAg NCs and H₂O molecules. i) The calculated contributions of atomic orbitals to the HOMO and LUMO of AuAg-D and AuAg-H-H₂O@Au NCs (please see **Figure R5**) reveal that where the contribution of O increases from 0.15% to 5.25%. This result supports that the lone pairs of O in the absorbed H₂O molecules have undergone orbital hybridization with the *d* atomic orbitals of the metal in the AuAg NCs through the formation of the Au-O bond. ii) However, the interaction of O in the H₂O molecule to the Au(Ag) atom in AuAg NCs is selective. As shown in **Figure R7**, it is found that the absorbed H₂O molecule preferentially interacts with the exposed Au atom in the Au₅Ag core ($E_{\text{Au-O}} = 0.00$ eV, **Figure R7a**) in the manner of forming Au-O bond. Furthermore, the absorbed H₂O molecule can also anchor to the Ag atom in the Au₅Ag core in the form of Ag-O bond ($E_{\text{Ag-O}} = 0.02$ eV, **Figure R7e**). Such interaction modes of the absorbed H₂O molecules are reasonable because the other four Au atoms in the Au₅Ag core are already bound to the motif through Au-S interaction, while the remaining Au and Ag atoms are bound only to metals. Accordingly, they have a relatively small coordination number which allows them to accommodate the absorbed H₂O molecules.

Figure R7. Structural optimization of the interaction position between the absorbed H₂O molecule and AuAg NCs. The relative energies of each model are given to gauge their corresponding thermodynamic stability.

2. *The authors also could give further insight into luminescence mechanism through*

DFT calculations, eg. using hole-electron analysis to investigate the characteristics for electronic excitation.

Reply: Thanks for your kind suggestion. We have carried out additional theoretical works to investigate their characteristics for electronic excitation transition through hole-electron analysis by using local excitation and charge transfer modes. i) The local excitation mode is recognized by the S/D value, where S indicates the calculated overlap integral of hole-electron distribution, and D refers to the calculated distance between the hole and electron centroids (*Nat. Commun.*, **2023**, 14, 3083). The larger S associated with the smaller D suggests the more evident local excitation in three AuAg NCs. The computed S/D values are 0.602, 1.263, and 0.674 for **AuAg-D**, **AuAg-H-H₂O@Au**, and **AuAg-H-H₂O@Ag** NCs, respectively, indicating that the absorbed H₂O molecules can promote the efficient local excitation, and therefore stronger PL intensity. ii) In addition, charge density difference (CDD) between the ground and the excited states is calculated to visualize the electron transfer within three AuAg NCs (**Figure R8**). To reveal the subtle changes due to the absorbed H₂O molecules, the distance of charge transfer (D_{CT}) based on the electron density variation during electron excitation has been calculated. The pristine **AuAg-D** NCs have a large D_{CT} value of 1.092 Å. However, the incorporation of H₂O molecules can remarkably shorten this value, especially for **AuAg-H-H₂O@Au** NCs, to 0.651 Å. The smaller D_{CT} value means that the holes and electrons are easier to recombine to give efficient PL, and accordingly results in significantly improved PLQY. All these data match the experimental results well. We have added necessary discussion in the main text to improve the scientific quality of this paper.

Figure R8. Quantitative hole-electron analysis of three AuAg NCs. a-c Computed charge density (CCD) difference between the ground and the excited states of the donor-accepter (D-A) pairs at an isovalue of 0.002 a.u. in **AuAg-D**, **AuAg-H-H₂O@Au**, and **AuAg-H-H₂O@Ag** NCs, respectively. The green and cyan represent an increase and decrease in electron density, respectively. d-f Quantitative charge-transfer analysis based on the atomic dipole corrected Hirshfeld (ADCH) atomic charges in **AuAg-D**, **AuAg-H-H₂O@Au**, and **AuAg-H-H₂O@Ag** NCs, respectively.

Revisions:

Page 21, Line 573-589:

“To give more theoretical insights into the PL mechanism of three AuAg NCs, we carried out hole-electron analysis to investigate their characteristics for electronic excitation transition, by using local excitation and charge transfer modes. According to the hole-electron theory, the dominant mode can be recognized by S/D value, where S indicates the calculated overlap integral of hole-electron distribution, and D refers to the calculated distance between the hole and electron centroids⁶⁷. The larger S associated with the smaller D suggests the more evident local excitation. The computed S/D values are 0.602, 1.263, and 0.674 for **AuAg-D**, **AuAg-H-H₂O@Au**, and **AuAg-H-H₂O@Ag** NCs, respectively, indicating that the absorbed H₂O molecules can promote the efficient local excitation. Charge density difference (CDD) between the ground and the excited states is calculated to visualize the electron transfer within three AuAg NCs (Supplementary Fig. 37). To reveal the subtle changes due to the absorbed H₂O molecules, the distance of charge transfer (D_{CT}) based on the *electron density variation* during electron excitation has been calculated. The pristine **AuAg-D** NCs have a large D_{CT} value of 1.092 Å. However, the incorporation of H₂O molecules can remarkably shorten this value, especially for **AuAg-H-H₂O@Au** NCs, to 0.651 Å. The smaller D_{CT} value means that the holes and electrons are easier to recombine to give efficient PL, and accordingly results in significantly improved PLQY.”

Page 29, Line 833-835:

“The electron and hole distributions of the models were constructed by Multiwfn and Visual Molecular Dynamics (VMD)⁷²⁻⁷⁴. The hole-electron analysis module of Multiwfn has been widely used to do the electron excitation analysis⁷⁵.”

Page 35, Line 1034-1036; Line 1044-1052:

Some necessary references have been cited in the theoretical calculation and discussion:

67. Qian, Y., Han, Y., Zhang, X., Yang, G., Zhang, G. & Jiang, H.-L. Computation-based regulation of excitonic effects in donor-acceptor covalent organic frameworks for enhanced photocatalysis. *Nat. Commun.* 14, 3083 (2023).
72. Lu, T. & Chen, W. Multiwfn: A multifunctional wavefunction analyzer. *J. Comput. Chem.* 33, 580–592 (2012).
73. Lu, T. A comprehensive electron wavefunction analysis toolbox for chemists, Multiwfn. *J. Chem. Phys.* 161, 082503 (2024).
74. Humphrey, W., Dalke, A. & Schulten, K. VMD: Visual molecular dynamics. *J. Mol. Graph.* 14, 33–38 (1996).
75. Liu, Z., Lu, T. & Chen, Q. An sp-hybridized all-carboatomic ring, cyclo[18]carbon: Electronic structure, electronic spectrum, and optical nonlinearity. *Carbon* 165, 461–467 (2020).

SI, Page 38, Figure S37:

Figure R8 has been included as **Figure S37** in the supporting information.

Replies to reviewers' comments and descriptions of revisions made

Comments by Reviewer #2:

The R1 manu is overall improved and publishable. The questions were addressed, in particular it's good to know that alcohols show a similar effect, though less than the effect of water. While Q1 needs more understanding, it's ok with me to publish R1 as is.

Reply: Thank you for your kind words and positive feedback. We greatly appreciate your time and effort in reviewing our manuscript and are pleased that our revised manuscript is publishable now. Your valuable insights have been instrumental in improving the quality of our work.

Comments by Reviewer #3:

The authors have answered all my questions, and I suggest publication as is.

Reply: Thank you for your positive feedback. We are delighted that the revised manuscript meets the standards of Nature Communications. Your constructive 1 comments and insights are invaluable in enhancing the quality of our work, and we sincerely appreciate your support throughout the review process.